# Information flows from hippocampus to auditory cortex during replay of verbal working memory items

Vasileios Dimakopoulos[1], Pierre Mégevand[2,3], Lennart H Stieglitz[1], Lukas Imbach[4,5], Johannes Sarnthein[1,5]*

[1]Klinik für Neurochirurgie, UniversitätsSpital Zürich, Universität Zürich, Zurich, Switzerland; [2]Département des neurosciences fondamentales, Faculté de médecine, Université de Genève, Genève, Switzerland; [3]Service de neurologie, Hôpitaux Universitaires de Genève, Geneva, Switzerland, Genève, Switzerland; [4]Schweizerisches Epilepsie Zentrum, Klinik Lengg AG, Zurich, Switzerland; [5]Neuroscience Center Zurich, ETH Zuric, Zurich, Switzerland

**Abstract** The maintenance of items in working memory (WM) relies on a widespread network of cortical areas and hippocampus where synchronization between electrophysiological recordings reflects functional coupling. We investigated the direction of information flow between auditory cortex and hippocampus while participants heard and then mentally replayed strings of letters in WM by activating their phonological loop. We recorded local field potentials from the hippocampus, reconstructed beamforming sources of scalp EEG, and – additionally in four participants – recorded from subdural cortical electrodes. When analyzing Granger causality, the information flow was from auditory cortex to hippocampus with a peak in the [4 8] Hz range while participants heard the letters. This flow was subsequently reversed during maintenance while participants maintained the letters in memory. The functional interaction between hippocampus and the cortex and the reversal of information flow provide a physiological basis for the encoding of memory items and their active replay during maintenance.

*For correspondence: johannes.sarnthein@usz.ch

Competing interest: The authors declare that no competing interests exist.

## Editor's evaluation

The work provides important information about the communication between the auditory cortex and hippocampus during phonological working memory. The results provide crucial insights into the networks involved in this fundamental process. The results are expected to be of broad interest to readers in the fields of working memory and cognitive neuroscience in general.

## Introduction

Working memory (WM) describes our capacity to represent sensory input for prospective use (*Baddeley, 2003*; *Christophel et al., 2017*). Maintaining content in WM requires communication within a widespread network of brain regions. The anatomical basis of WM was shown noninvasively with EEG/MEG (*Michels et al., 2008*; *Sarnthein et al., 1998*; *Tuladhar et al., 2007*; *Polanía et al., 2012*; *Näpflin et al., 2008*; *Bidelman et al., 2021*; *Pavlov and Kotchoubey, 2022*; *Hsieh and Ranganath, 2014*) and invasively with intracranial local field potentials (LFP; *Cogan et al., 2017*; *Raghavachari et al., 2001*; *Rizzuto et al., 2003*; *Maris et al., 2011*; *van Vugt et al., 2010*; *Leszczyński et al., 2015*; *Johnson et al., 2018a*; *Johnson et al., 2018b*; *Boran et al., 2019*; *Li et al., 2022*;

**eLife digest** Every day, the brain's ability to temporarily store and recall information – called working memory – enables us to reason, solve complex problems or to speak. Holding pieces of information in working memory for short periods of times is a skill that relies on communication between neural circuits that span several areas of the brain. The hippocampus, a seahorse-shaped area at the centre of the brain, is well-known for its role in learning and memory. Less clear, however, is how brain regions that process sensory inputs, including visual stimuli and sounds, contribute to working memory.

To investigate, Dimakopoulos et al. studied the flow of information between the hippocampus and the auditory cortex, which processes sound. To do so, various types of electrodes were placed on the scalp or surgically implanted in the brains of people with drug-resistant epilepsy. These electrodes measured the brain activity of participants as they read, heard and then mentally replayed strings of up to 8 letters. The electrical signals analysed reflected the flow of information between brain areas.

When participants read and heard the sequence of letters, brain signals flowed from the auditory cortex to the hippocampus. The flow of electrical activity was reversed while participants recalled the letters. This pattern was found only in the left side of the brain, as expected for a language related task, and only if participants recalled the letters correctly.

This work by Dimakopoulos et al. provides the first evidence of bidirectional communication between brain areas that are active when people memorise and recall information from their working memory. In doing so, it provides a physiological basis for how the brain encodes and replays information stored in working memory, which evidently relies on the interplay between the hippocampus and sensory cortex.

*Schwiedrzik et al., 2018*) and single-unit recordings (*Boran et al., 2019*; *Schwiedrzik et al., 2018*; *Kamiński et al., 2017*; *Kornblith et al., 2017*; *Rutishauser et al., 2021*).

In cortical brain regions, WM maintenance correlates with sustained neuronal oscillations, most frequently reported in the theta-alpha range ([4 12] Hz; *Michels et al., 2008*; *Sarnthein et al., 1998*; *Tuladhar et al., 2007*; *Polanía et al., 2012*; *Näpflin et al., 2008*; *Pavlov and Kotchoubey, 2022*; *Hsieh and Ranganath, 2014*; *Cogan et al., 2017*; *Raghavachari et al., 2001*; *Rizzuto et al., 2003*; *Maris et al., 2011*; *van Vugt et al., 2010*; *Leszczyński et al., 2015*; *Johnson et al., 2018a*; *Johnson et al., 2018b*; *Boran et al., 2019*; *Li et al., 2022*) or at even lower frequencies (*Kumar et al., 2021*; *Rezayat et al., 2021*). Also in the hippocampus, WM maintenance was associated with sustained theta-alpha oscillations (*van Vugt et al., 2010*; *Boran et al., 2019*). As a hallmark for WM maintenance, persistent neuronal firing was reported during the absence of sensory input, indicating the involvement of the medial temporal lobe in WM (*Boran et al., 2019*; *Kamiński et al., 2017*; *Kornblith et al., 2017*; *Boran et al., 2022*).

At the network level, synchronized oscillations have been proposed as a mechanism for functional interactions between brain regions (*Fries, 2015*; *Pesaran et al., 2018*). It is thought that these oscillations show temporal coupling of the low-frequency phase for long-range communication between cortical areas (*Sarnthein et al., 1998*; *Polanía et al., 2012*; *Maris et al., 2011*; *Johnson et al., 2018a*; *Johnson et al., 2018b*; *Boran et al., 2019*; *Solomon et al., 2017*). This synchronization suggests an active maintenance process through reverberating signals between brain regions.

We here extend previous studies with the same task (*Michels et al., 2008*; *Boran et al., 2019*) by recording from four participants with hippocampal LFP and direct cortical recordings (ECoG) from electrodes over primary auditory, parietal, and occipital cortical areas. Given the low incidence of the epileptogenic zone in parietal cortex, parietal ECoG recordings are rare. To benefit from the wide spatial coverage of scalp EEG, we analyzed the directed functional coupling between hippocampal LFP and the beamforming sources of scalp EEG in all 15 participants. We found that the information flow was from auditory cortex to hippocampus during the encoding of WM items, and the flow was from hippocampus to auditory cortex for the replay of the items during the maintenance period.

# Results

## Task and behavior

Fifteen participants (median age 29 years, range [18–56], 7 male, *Table 1*) performed a modified Sternberg WM task (71 sessions in total, 50 trials each). In the task, items were presented all at once rather than sequentially, thus separating the encoding period from the maintenance period. In each trial, the participant was instructed to memorize a set of 4, 6, or 8 letters presented for 2 s (encoding). The number of letters was thus specific for the memory workload. The participants read the letters themselves and heard them spoken at the same time. Since participants had difficulties reading eight letters within the 2 s encoding period, also hearing the letters assured their good performance. After a delay (maintenance) period of 3 s, a probe letter prompted the participant to retrieve their memory (retrieval) and to indicate by button press ('IN' or 'OUT') whether or not the probe letter was a member of the letter set held in memory (*Figure 1a*). During the maintenance period, participants rehearsed the verbal representation of the letter strings subvocally, i.e., mentally replayed the memory items. Participants had been instructed to employ this strategy, and they confirmed after the sessions that they had indeed employed this strategy. This activation of the phonological loop (*Baddeley, 2003*) is a component of verbal WM as it serves to produce an appropriate behavioral response (*Christophel et al., 2017*).

The mean correct response rate was 91% (both for IN and OUT trials). The rate of correct responses decreased with set size from a set size of 4 (97% correct responses) to set sizes of 6 (89%) and 8 (83%) (*Figure 1b*). Across the participants, the memory capacity averaged 6.1 (Cowan's K, [correct IN rate +correct OUT rate −1]×set size), which indicates that the participants were able to maintain at least six letters in memory. The mean response time (RT) for correct trials (3045 trials) was 1.1±0.5 s and increased with workload from set size 4 (1.1±0.5 s) to 6 (1.2±0.5 s) and 8 (1.3±0.6 s), 53 ms/item (*Figure 1c*). Correct IN/OUT decisions were made more rapidly than incorrect decisions (1.1±0.5 s vs 1.3±0.6 s). These data show that the participants performed well in the task and that the difficulty of the trials increased with the number of letters in the set. In further analysis, we focused on correct trials with set size 6 and 8 letters to assure hippocampal activation and hippocampo-cortical interaction as shown earlier (*Boran et al., 2019*).

## Power spectral density in cortical and hippocampal recordings

To investigate how cortical and hippocampal activity subserves WM processing, we analyzed the LFP recorded in the hippocampus (*Figure 1d*, *Figure 1—figure supplement 1*, *Supplementary file 1*) together with ECoG from cortical strip electrodes (*Figure 2a*, *Figure 3a and f*). In the following, we present power spectral density (PSD) time-frequency maps from representative electrode contacts. In an occipital recording of Participant 1 (grid contact H3), strong gamma activity (>40 Hz) in the relative PSD occurred while the participant viewed the letters during encoding (increase >100% with respect to fixation, *Figure 2b*). Similarly, encoding elicited gamma activity in a temporal recording over auditory cortex (increase >100%, grid contact C2, *Figure 2c*), similar as in *Kumar et al., 2021*. Gamma increased significantly only in temporal and occipital-parietal contacts (permutation test with z-score >1.96, *Figure 2a*).

After the letters disappeared from the screen, activity occurred in the [11 14] Hz range (high alpha/low beta, *Figure 2b*) toward the end of the maintenance period in temporal and occipital contacts (permutation test p<0.05, *Figure 2d*). Similarly, the temporal scalp EEG of Participant 2 (black rimmed disk denotes electrode site T3 in *Figure 3a*) showed activity during encoding and maintenance, albeit at lower frequencies (*Figure 3b*); this pattern was found only in scalp EEG and not in ECoG, probably because the strip electrode was not located over auditory cortex. In Participant 3, a similar pattern occurred in the PSD of a temporo-parietal recording (most posterior strip electrode contact, *Figure 3f*), where the appearance of the letters prompted gamma activity and the maintenance period showed alpha activity ([8 11] Hz, *Figure 3g*). Similarly, in the electrode contacts on right parietal cortex of Participant 4 (*Figure 3k*), the letter stimulus elicited gamma activity and the maintenance period showed alpha activity (8–11 Hz, *Figure 3l*).

The site of the participants' maintenance activity coincides with the generator of scalp EEG that was found in the parietal cortex for the same task (*Michels et al., 2008*). The PSD thereby confirmed the findings of local synchronization of cortical activity during WM maintenance (*Michels et al., 2008*; *Bidelman et al., 2021*; *Pavlov and Kotchoubey, 2022*).

**Table 1.** Participant characteristics and results of Granger causality analysis.

For each participant, we report the atlas parcels that contained EEG sources with the maximal t-value and the t-value of sources in auditory cortex (Heschl gyrus) during encoding and maintenance (non-parametric cluster-based permutation test p<0.05). In each participant, the vast majority of the significant LCMV sources were in the left hemisphere, both during encoding (≥87%) and during maintenance (≥81%). We also report the net information flow (ΔGranger) for correct and incorrect trials in the direction auditory cortex → hippocampus during encoding and in the direction hippocampus → auditory cortex during maintenance.

| Participant | Pathology | Encoding | | | | | | Maintenance | | | | | |
| --- | --- | --- | --- | --- | --- | --- | --- | --- | --- | --- | --- | --- | --- |
| | | Maximal LCMV source | Significant LCMV sources in the left hemisphere (%) | max. t-value | Heschl t-value | Heschl ΔGranger correct trials | Heschl ΔGranger incorrect trials | Maximal LCMV source | Significant LCMV sources in the left hemisphere (%) | max. t-value | Heschl t-value | Heschl ΔGranger correct trials | Heschl ΔGranger incorrect trials |
| 1 | hippocampal sclerosis | Heschl / Temporal Inferior L | 100 | 17.8 | 17.8 | −0.036 | 0.087 | Frontal Mid Orb / Heschl L | 100 | 10.1 | 10.1 | 0.025 | −0.037 |
| 2 | non-lesional | Heschl L | 100 | 19.8 | 19.8 | −0.017 | 0.016 | Temporal Inferior L | 96 | 11.4 | 10 | 0.093 | 0.002 |
| 3 | focal cortical dysplasia | Temporal Superior L | 91 | 24.2 | 16.3 | −0.060 | −0.013 | Heschl L | 91 | 14.6 | 14.6 | 0.065 | 0.005 |
| 4 | unclear etiology | Frontal Inferior L | 100 | 18.3 | 16.6 | −0.006 | 0.003 | Heschl L | 100 | 13.4 | 13.4 | 0.035 | −0.002 |
| 5 | brain contusion | Temporal Superior L | 100 | 6.9 | 5.4 | −0.003 | −0.002 | Heschl L | 96 | 7.6 | 7.6 | 0.021 | 0.048 |
| 6 | hippocampal sclerosis | Supramarginal L | 98 | 11.1 | 9.7 | −0.049 | 0.025 | Temporal Pole Superior L | 93 | 19.8 | 17.9 | 0.039 | −0.024 |
| 7 | xanthoastrozytoma | Lingual R | 87 | 12.9 | 11.5 | −0.059 | 0.042 | Caudate L | 85 | 9.3 | 8.2 | 0.042 | −0.036 |
| 8 | focal cortical dysplasia | Caudate L | 100 | 18.8 | 16.2 | −0.040 | 0.013 | Parietal Superior L | 100 | 9.9 | 8.3 | 0.017 | −0.031 |
| 9 | gliosis | Cingulum Anterior L | 100 | 12.3 | 11.2 | −0.051 | 0.012 | Parietal Inferior L | 100 | 12.6 | 11.2 | 0.050 | −0.015 |
| 10 | hippocampal sclerosis | Heschl L | 100 | 7.9 | 7.9 | −0.070 | −0.016 | Cingulum Mid L | 100 | 2.7 | 2.3 | 0.018 | −0.081 |
| 11 | hippocampal sclerosis | Cingulum Anterior L | 100 | 10.8 | 6.3 | −0.052 | −0.011 | Cuneus L | 86 | 4.8 | 4.3 | 0.020 | −0.062 |
| 12 | hippocampal sclerosis | Temporal Superior L | 95 | 8.1 | 6.7 | −0.022 | 0.000 | Temporal Pole Mid L | 100 | 7.3 | 4.9 | 0.012 | 0.003 |
| 13 | hippocampal sclerosis | Temporal Superior R | 93 | 10.8 | 6.9 | −0.018 | 0.019 | Parietal Superior R | 81 | 5.6 | 4.9 | 0.020 | −0.022 |
| 14 | hippocampal sclerosis | Temporal Superior L | 100 | 16.8 | 13.3 | −0.053 | 0.022 | Heschl L | 100 | 11.8 | 11.8 | 0.085 | −0.089 |
| 15 | hippocampal sclerosis | Heschl L | 98 | 9.1 | 9.1 | −0.061 | −0.005 | Heschl L | 100 | 10.4 | 10.4 | 0.069 | −0.004 |

LCMV, linearly constrained minimum variance; ΔGranger, difference of GC spectra.

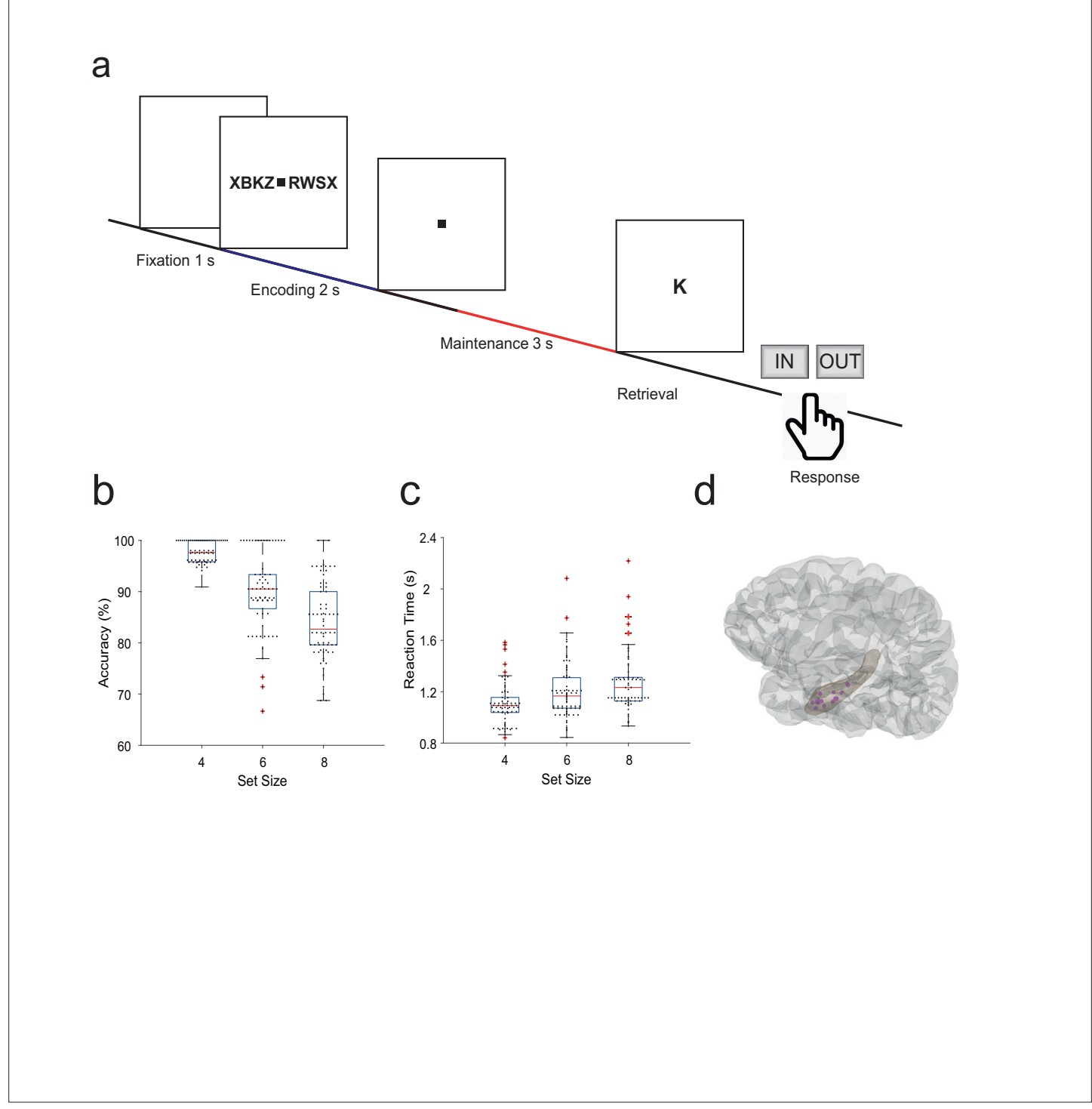

**Figure 1.** Task and recording sites. (**a**) In the task, sets of consonants are presented and have to be memorized. The set size (4, 6, or 8 letters) determines working memory workload. In each trial, presentation of a letter string (encoding period, 2 s) is followed by a delay (maintenance period, 3 s). After the delay, a probe letter is presented. Participants indicate whether the probe was in the letter string or not. (**b**) Response accuracy decreases with set size (71 sessions). (**c**) Reaction time increases with set size (53 ms/item). (**d**) The tip locations of the hippocampal local field potentials electrodes for all participants (N=15) are projected in a hippocampal surface.

The online version of this article includes the following figure supplement(s) for figure 1:

**Figure supplement 1.** Hippocampal contact locations.

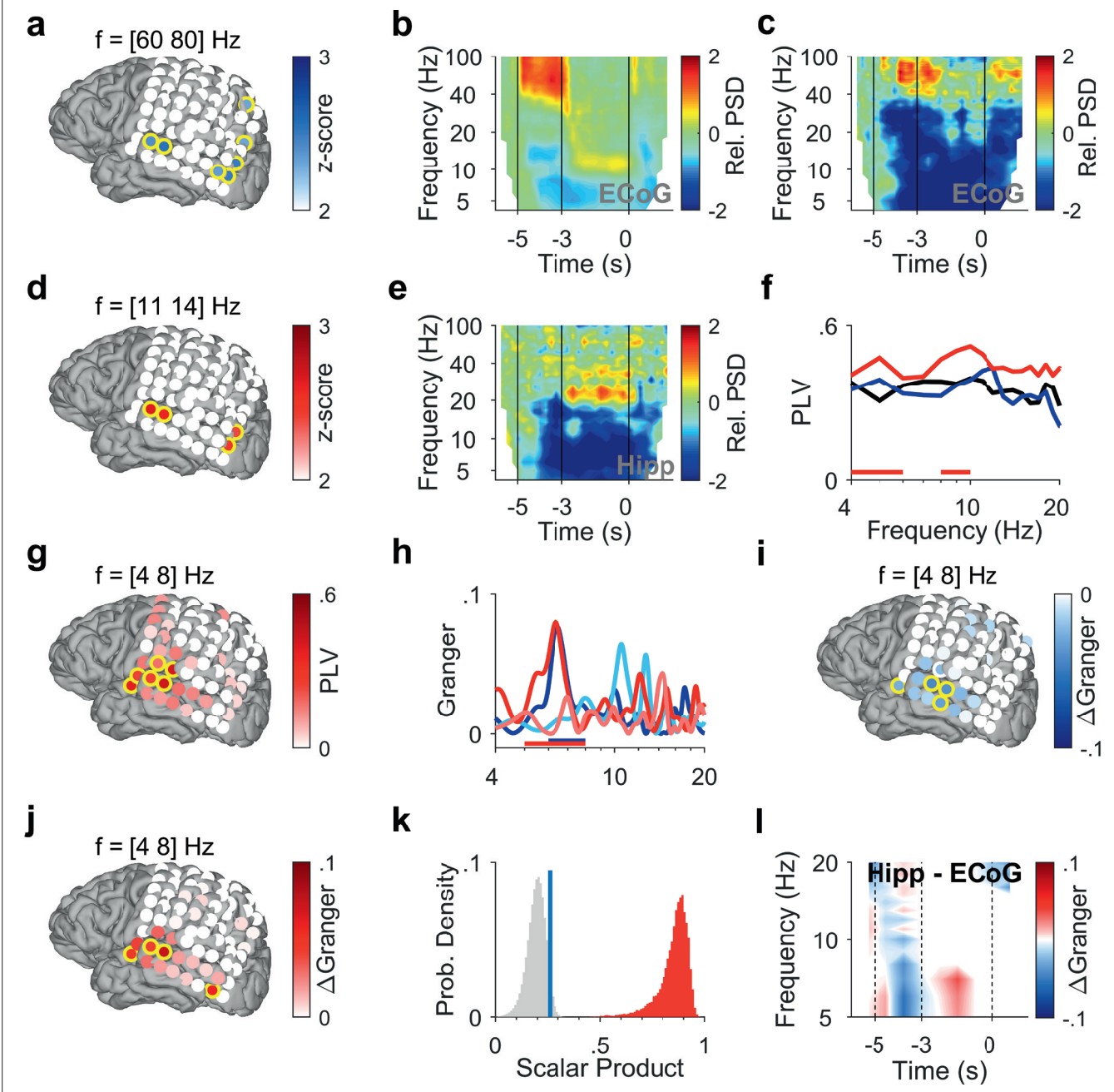

**Figure 2.** Encoding and replay of letters in Participant 1. (**a**) Location of the ECoG contacts over temporal and parietal cortex for Participant 1. Relative gamma power spectral density (PSD; [60 80] Hz) during encoding ([−3.5 −3] s) is maximal for contacts over temporal and occipital-parietal cortex. (**b**) The relative PSD in the occipital contact (contact H3) over visual cortex shows gamma activity (>40 Hz) during encoding ([−5 −3] s) while the subject sees and hears the letters. Sustained low beta activity ([11 14] Hz) appears toward the end of the maintenance period ([−3 0] s). (**c**) The relative PSD in the temporal contact (contact C2) over auditory cortex shows gamma activity ([60 80] Hz) during the last second of encoding ([−4 −3] s) while the subject sees and hears the letters. (**d**) Relative beta PSD ([11 14] Hz) during maintenance ([−2 0] s) is maximal for contacts over temporal and occipital cortex. (**e**) Hippocampal PSD shows sustained beta activity toward the end of maintenance. (**f**) Phase-locking value (PLV) between hippocampus and auditory cortex (contact C3) during fixation (black), encoding (blue), and maintenance (red). The PLV spectra show a broad frequency distribution. The PLV during maintenance is higher than during fixation. Red bars: frequency ranges of significant PLV difference (p<0.05, cluster-based non-parametric permutation test against a null distribution with scrambled trials during fixation and maintenance). (**g**) PLV between hippocampus and cortex in theta ([4 8] Hz) during maintenance ([−2 0] s) is highest to contacts over auditory cortex. (**h**) Spectral Granger causality. During encoding ([−5 −3] s), auditory cortex (contact C2) predicts hippocampus ( [6 8] Hz, dark blue curve exceeds light blue curve). During maintenance ([−2 0] s), hippocampus predicts auditory cortex ( [5 8] Hz, dark red curve exceeds light red curve). Bars: frequency range of significant ΔGranger (p<0.05, cluster-based non-parametric permutation test against a null distribution with scrambled trials during encoding (blue) and maintenance (red). (**i**) Net information flow ΔGranger ([4

*Figure 2 continued on next page*

*Figure 2 continued*

8] Hz) during encoding ([−5 −3] s). ECoG over auditory cortex predicts hippocampal local field potentials. (**j**) Net information flow ΔGranger ( [4 8] Hz) during maintenance ([−2 0] s). Hippocampus is maximal in predicting auditory cortex (contact C2 and surrounding contacts). (**k**) Statistical significance of the spatial spread of contacts with high ΔGranger ([4 8] Hz) during maintenance ([−2 0] s). We calculated the scalar product between two spread vectors. We then tested the statistical significance of the scalar product. The true distribution (red) is clearly distinct from the null distribution (gray, blue bar marks 95th percentile). (**l**) The Granger time-frequency map illustrates the time course of the spectra of panel (h). During encoding, net information (ΔGranger) flows from auditory cortex to hippocampus (blue). During maintenance, the information flow is reversed from hippocampus to auditory cortex (red) indicating the replay of letters in memory. Grid contacts with significant increase are marked with a yellow rim (p<0.05, cluster-based non-parametric permutation test against a null distribution with scrambled trials). The time course in time-frequency maps is shown relative to the fixation period (**b, c, e**). Colors of Granger spectra indicate information flow: dark blue, cortex to hippocampus during encoding; light blue, hippocampus to cortex during encoding; dark red, hippocampus to cortex during maintenance; light red, cortex to hippocampus during maintenance. ΔGranger is the difference between spectra, where ΔGranger <0 denotes information flow cortex→hippocampus and ΔGranger >0 denotes information flow hippocampus→cortex. Grid contacts are identified by column (anterior A to posterior H) and row (inferior 1 to superior 8).

In the hippocampus of all four participants, we found elevated activity in the beta range ([12 24] Hz) toward the end of the maintenance period (increase >100%, **Figure 2e**, **Figure 3c, h and m**), confirming the hippocampal contribution to processing of this task (**Boran et al., 2019**).

## Functional coupling between hippocampus and cortex

To investigate the functional coupling between cortex and hippocampus, we first calculated the phase-locking value (PLV). In Participant 1, we found high PLV over a broad frequency range in contacts over auditory cortex throughout the trial. Compared to encoding, maintenance showed enhanced PLV in the theta range between hippocampal LFP and cortical ECoG (PLV=0.4 in contact C3, permutation test p<0.05, **Figure 2f**). PLV in the [4 8] Hz theta range increased significantly with several contacts over auditory cortex (permutation test p<0.05, **Figure 2g**). This speaks for a functional coupling between auditory cortex and hippocampus mediated by synchronized oscillations (**Rezayat et al., 2021**).

## Directed functional coupling between hippocampus and ECoG

What was the directionality of the information flow during encoding and maintenance in a trial? We used spectral Granger causality (GC) as a measure of directed functional connectivity to determine the direction of the information flow between auditory cortex and hippocampus in Participant 1 during the trials. During encoding, the information flow was from auditory cortex to hippocampus with a maximum in the theta frequency range (dark blue curve in **Figure 2h**). The net information flow ΔGranger (GC hipp→cortex – GC cortex→hipp) during encoding was significant in the [6 8] Hz range (blue bar in **Figure 2h**, p<0.05 permutation test against a null distribution). During maintenance, the information flow in the theta frequency range was reversed (dark red curve), i.e., from hippocampus to auditory cortex (dark red curve in **Figure 2h**). The net information flow ΔGranger during maintenance was significant in the [5 8] Hz range (red bar in **Figure 2h**, p<0.05 permutation test against a null distribution). Concerning the spatial spread of the theta GC, the maximal net information flow ΔGranger (GC hipp→cortex – GC cortex→hipp) during encoding occurred from auditory cortex to hippocampus (p<0.05, permutation test, **Figure 2i**). During maintenance, the theta ΔGranger was significant from hippocampus to both auditory cortex and occipital cortex (permutation test p<0.05, **Figure 2j**). Interestingly, in Participant 1, the distribution of high ΔGranger coincides with the distribution of high PLV: both show a spatial maximum to grid contacts over auditory cortex and both appear in the theta frequency range.

We next tested the statistical significance of the spatial spread of contacts with high ΔGranger ([4 8] Hz) during maintenance ([−2 0] s). To provide a sound statistical basis, we tested the spatial distribution of GC on the grid contacts against a null distribution. The activation on grid contacts was reshaped into a grid vector. The spatial collinearity of two grid vectors was captured by their scalar product. We next performed 200 iterations of random trial permutations. For each iteration, we selected two subsets of trials, and we calculated the scalar product between the two vectors corresponding to these subsets. We then tested the statistical significance of the scalar product (**Figure 2k**). The true distribution (red) is clearly distinct from the null distribution (gray, blue bar marks the 95th percentile). The analogous procedure was performed for PSD (**Figure 2a and d**), PLV (**Figure 2g**), and GC during encoding (**Figure 2i**), which gave equally significant results in all cases.

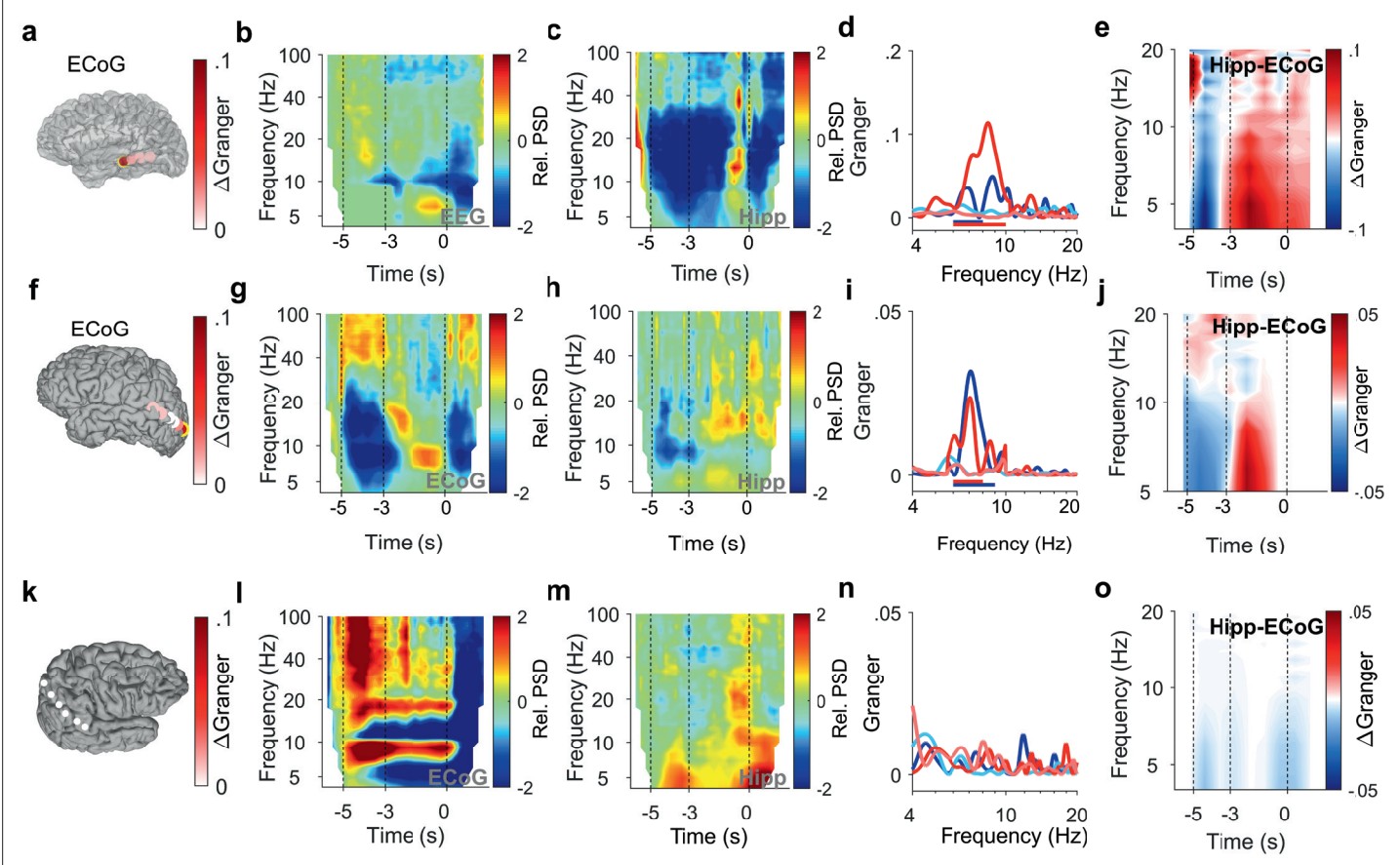

**Figure 3.** Encoding and replay of letters in three participants with ECoG. (**a**) Location of the ECoG contacts in Participant 2. The most anterior strip contact records from auditory cortex. Color bar: ΔGranger during maintenance ([4 8] Hz). (**b**) The relative power spectral density (PSD) in the temporal scalp EEG electrode (T5) shows beta activity ([14 25] Hz) during encoding ([−5 −3] s) while the subject sees and hears the letters. Sustained theta activity ([6 9] Hz) appears toward the end of the maintenance period ([−3 0] s). (**c**) Hippocampal PSD shows alpha-beta activity (9–18 Hz) toward the end of maintenance. (**d**) Spectral Granger causality (GC). During encoding, the auditory cortex predicts hippocampus ([6 8] Hz, dark blue curve exceeds light blue curve). During maintenance, hippocampal local field potentials (LFP) predict auditory cortex ([6 10] Hz, dark red curve exceeds light red curve). (**e**) The time-frequency map illustrates the time course of ΔGranger in Participant 2. (**f**) Location of the ECoG contacts in Participant 3. The most posterior contact records from visual cortex (yellow rimmed disk). Color bar: ΔGranger during maintenance ([4 8] Hz). (**g**) The relative PSD in the most posterior contact (yellow rimmed disk, panel (f)) shows gamma during encoding while the subject sees the letters. Sustained alpha activity ([8 11] Hz) appears toward the end of the maintenance period. (**h**) Hippocampal PSD shows sustained beta activity ([13 21] Hz) toward the end of the maintenance. (**i**) Spectral GC. During encoding, the occipital ECoG predicts hippocampus (6–9 Hz, dark blue curve exceeds light blue curve). During maintenance, hippocampal LFP predicts ECoG ([6 8] Hz, dark red curve exceeds light red curve). (**j**) The time-frequency map illustrates the time course of ΔGranger in Participant 3. (**k**) Location of the ECoG contacts in Participant 4 on right parietal cortex. Color bar: ΔGranger during maintenance ([4 8] Hz). (**l**) The relative PSD in contact over the right parietal lobule shows gamma during encoding while the subject sees the letters. Sustained alpha activity ([8 11] Hz) appears during the maintenance period. (**m**) Hippocampal PSD shows sustained beta activity ([13 21] Hz) toward the end of the maintenance. (**n**) Spectral GC. Task performance does not elicit significant GC to the right parietal cortex in Participant 4. (**o**) The time-frequency map illustrates the time course of ΔGranger in Participant 4. Task performance does not elicit significant GC to the right parietal cortex in Participant 4. Color bar: ΔGranger during maintenance ([4 8] Hz). Grid contacts with significant increase in ΔGranger are marked with a yellow rim (permutation test p<0.05). The time course in time-frequency maps is shown relative to the fixation period (**b, c, g, h, l,m**). Colors of Granger spectra indicate information flow: dark blue, cortex to hippocampus during encoding; light blue, hippocampus to cortex during encoding; dark red, hippocampus to cortex during maintenance; light red, cortex to hippocampus during maintenance. ΔGranger is the difference between spectra where ΔGranger <0 denotes information flow cortex→hippocampus and ΔGranger >0 denotes information flow hippocampus→cortex. Bars: frequency range of significant ΔGranger (p<0.05), cluster-based non-parametric permutation test against a null distribution with scrambled trials during encoding and maintenance, respectively.

As a further illustration of the ΔGranger time course, the time-frequency plot (*Figure 2I*) shows the difference between GC spectra (GC hipp→cortex – GC cortex→hipp) at each time point, where blue indicates net flow from auditory cortex to hippocampus and red indicates net flow from hippocampus to auditory cortex.

Similarly in Participant 2, the time course of GC followed the same pattern between auditory cortex (anterior strip electrode contact in *Figure 3a*) and hippocampus (*Figure 3d and e*). Among the participants that had both LFP and temporo-parietal ECoG recordings, Participant 3 had an electrode contact over left visual cortex; the sensory localization was indexed by the strong gamma activity in the most posterior contact of the strip electrode (*Figure 3g*). The time course of information flow between visual cortex and hippocampus (*Figure 3i and j*) followed the same pattern as described for the auditory cortex above. Interestingly, the pattern appeared with LFP recorded from right hippocampus in Participant 3 (*Supplementary file 1*). However, in Participant 4, the recordings from the right cortical hemisphere (*Figure 3k*) did not show significant GC between LFP and ECoG during task performance (*Figure 3n and o*).

Thus, we showed in recordings from the left cortical hemisphere that letters were encoded with information flow from sensory cortex to hippocampus; conversely, the information flow from hippocampus to sensory cortex indicated the replay of letters during maintenance.

## Source reconstruction of the scalp EEG

We used beamforming (*Oostenveld et al., 2011*) to reconstruct the EEG sources during encoding and maintenance for each of the 15 participants (*Table 1*). We tested whether the sources during fixation differed from sources during encoding and during maintenance (non-parametric cluster-based permutation t-test *Maris and Oostenveld, 2007*; *Popov et al., 2018*). In each participant, the proportion of significant sources in the left hemisphere exceeded 80% of all significant sources. Across all participants, the spatial activity pattern during both encoding and maintenance showed the highest significance in frontal and temporal areas of the left hemisphere (*Figure 4—figure supplement 1*).

## Directed functional coupling between hippocampus and averaged EEG sources

The synchronization between hippocampal LFP and EEG sources (N=15 participants) confirmed the directed functional coupling found in the three participants with ECoG. We first calculated the GC between hippocampus and the EEG beamforming sources in the auditory cortex. We found that the mean GC spectra resembled the GC spectrum for ECoG in the theta frequency range ([4 8] Hz, *Figure 4a*). During encoding, the net information flow was from auditory cortex to hippocampus (light blue curve – dark blue curve, blue bar, p<0.05, group cluster-based permutation test). During maintenance, the net information flow was reversed (dark red curve – light red curve, red bar, p<0.05, group cluster-based permutation test), i.e., from hippocampus to auditory cortex. Interestingly, the pattern appeared with LFP recorded from the right hippocampus in several participants (*Supplementary file 1*). A similar GC pattern emerged when using the signals from left temporal scalp electrodes but was eliminated when using a Laplacian derivation. Thus, both for ECoG and EEG beamforming sources, GC showed the same bidirectional effect in theta between auditory cortex and hippocampus.

To explore the spatial distribution, we computed GC also for other areas of cortex. We averaged the net information flow (ΔGranger) in the theta range across the participants and projected it onto the inflated brain surface (*Figure 4b and c*). During encoding, the mean information flow was strongest from auditory cortex to hippocampus (ΔGranger=−0.049, p=0.0009, Kruskal-Wallis test, *Figure 4b*). For all other areas, the mean ΔGranger was also from cortex to hippocampus but the effect was weaker (mean ΔGranger = [–0.03 0], Dunn's test, Bonferroni corrected). During maintenance (*Figure 4c*) the information flow was reversed. While all areas had information flow from hippocampus to cortex (ΔGranger = [0.02], Dunn's test, Bonferroni corrected), the strongest flow appeared from hippocampus to auditory cortex (ΔGranger=0.034, p=0.001, Kruskal-Wallis test).

## Directed functional coupling and the participants' performance

The reversal of ΔGranger appeared in all 15 participants individually (*Figure 4d*). We averaged ΔGranger for each participant in the [4 8] Hz theta frequency range. The ΔGranger between hippocampus and auditory cortex, was negative during encoding and was positive during maintenance in the theta frequency range (p=4.1e-10, paired permutation test). The directionality and its reversal was missing for all other areas, e.g., lateral prefrontal cortex (p=0.16, paired permutation test, *Figure 4e*). Of note, all analyses up to here were performed on correct trials only.

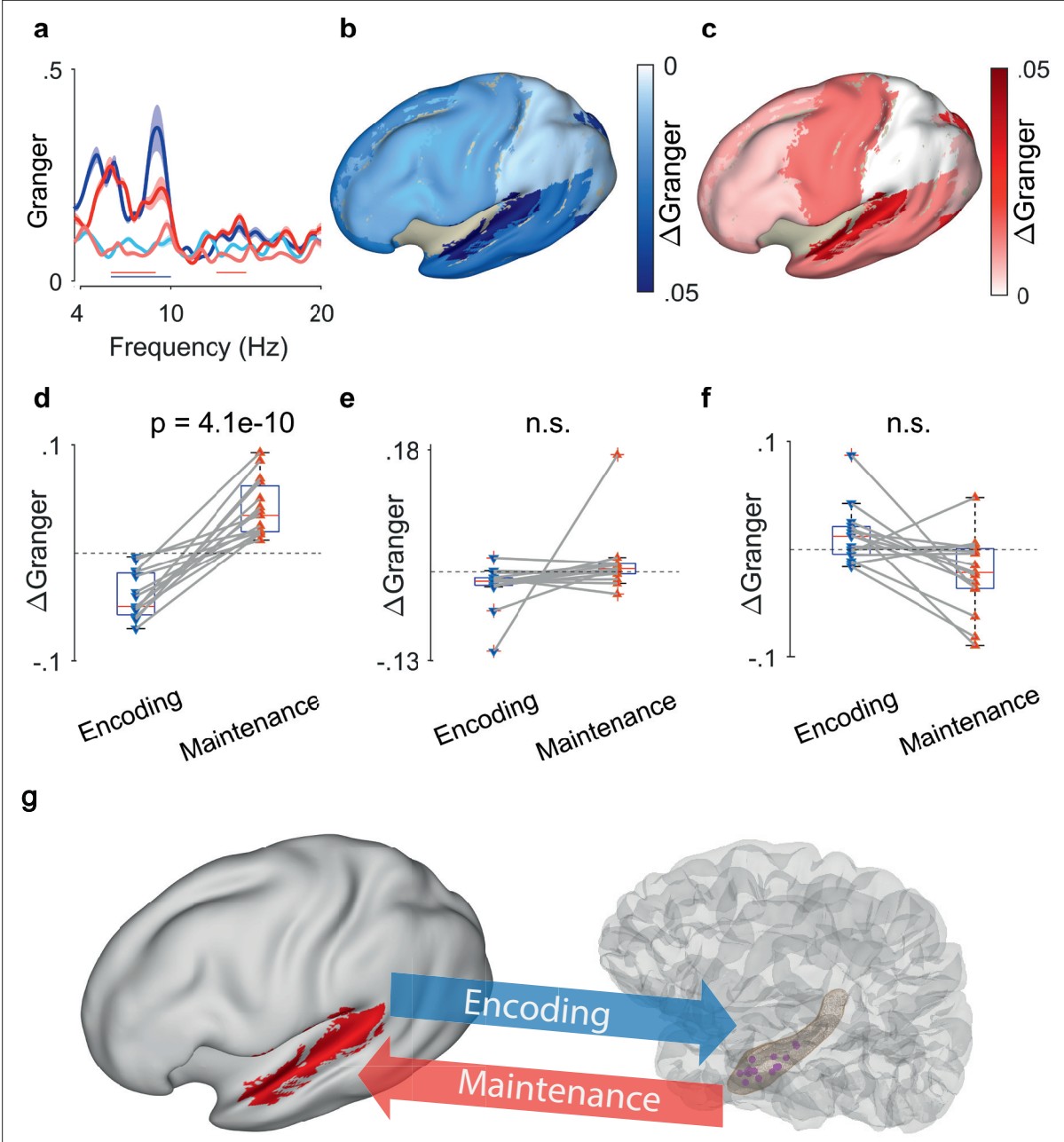

**Figure 4.** Granger causality (GC) between hippocampal local field potentials (LFP) and EEG sources. (**a**) Spectral GC between hippocampal LFP and auditory EEG sources, averaged over all N=15 participants. The shaded area indicates the variability across the population. During encoding, the net Granger (ΔGranger) indicates information flow from auditory cortex to hippocampus ([6 10] Hz, blue bar). During maintenance, ΔGranger indicates information flow from hippocampal LFP to auditory cortex (red bars, [6 9] Hz, [13 15] Hz). Bars: frequency range of significant ΔGranger (p<0.05), group cluster-based non-parametric permutation t-test against a null distribution with scrambled trials during encoding and maintenance. Colors of Granger spectra indicate information flow: dark blue, cortex to hippocampus during encoding; light blue, hippocampus to cortex during encoding; dark red, hippocampus to cortex during maintenance; light red, cortex to hippocampus during maintenance. (**b**) The median net information flow (ΔGranger) in the [4 8] Hz range during encoding is projected onto an inflated brain surface. The maximal ΔGranger appeared from temporal superior gyrus (median ΔGranger=–0.049) indicating information flow from auditory cortex to hippocampus. Negative values of median ΔGranger appeared also in other areas, albeit less intense (ΔGranger>–0.03). (**c**) The median net information flow (ΔGranger) in the [4 8] Hz range during maintenance is projected onto an inflated brain surface. The maximal ΔGranger appeared from temporal superior gyrus (median ΔGranger=0.034) indicating an information flow from hippocampus to auditory cortex. Positive values of median ΔGranger appeared also in other areas, albeit less intense (ΔGranger <0.02). (**d**) The maximal ΔGranger in the [4 8] Hz range was negative during encoding (blue, auditory cortex → hippocampus, median ΔGranger=–0.049) and positive during maintenance (red, hippocampus → auditory cortex, median ΔGranger=0.034) for each participant (red and blue connected marker, paired permutation

*Figure 4 continued on next page*

*Figure 4 continued*

test, correct trials only). The mean values and statistical significance derive only from 10% of the correct trials in order to balance the number of incorrect trials. (**e**) The net information flow between hippocampal LFP and lateral prefrontal cortex in the [4 8] Hz range has a lower median than to auditory cortex and higher variability (correct trials only, p=0.16, paired permutation test, not significant). (**f**) For incorrect trials, the maximal ΔGranger in the [4 8] Hz range is highly variable (p=0.37, paired permutation test, not significant). (**g**) Bidirectional information flow between cortical sites and hippocampus in the working memory network. The GC analysis suggests a surprisingly simple model of information flow during the task. During encoding, letter strings are verbalized as subvocal speech; the incoming information flows from auditory cortex to hippocampus. During maintenance, participants actively recall and rehearse the subvocal speech in the phonological loop; GC indicates an information flow from hippocampus to cortex as the physiological basis for the replay of the memory items.

The online version of this article includes the following figure supplement(s) for figure 4:

**Figure supplement 1.** Spatial activation pattern of EEG beamforming sources.

Finally, we established a link between the participants' performance and ΔGranger. For incorrect trials, the net information flow ΔGranger from auditory cortex to hippocampus did not show the same directionality in all participants and did not reverse in direction (p=0.37, paired permutation test, *Figure 4f*). Since participants performed well (median performance 91%), we balanced the numbers of correct and incorrect trials. We calculated the GC in a subset of correct trials (median of 200 permutations of a number of correct trials that equals the mean percentage of incorrect trials=10%); the effect was equally present for the subset of correct trials (p<0.0005). This suggests that timely information flow, as indexed by GC, is relevant for producing a correct response.

## Discussion

WM describes our capacity to represent sensory input for prospective use. Our findings suggest that this cognitive function is subserved by bidirectional oscillatory interactions between memory neurons in the hippocampus and sensory neurons in the auditory cortex as indicated by phase synchrony and GC. In our verbal WM task, the encoding of letter items is isolated from the maintenance period in which the active rehearsal of memory items is central to achieve correct performance. First, analysis of task-induced power showed sustained oscillatory activity in cortical and hippocampal sites during the maintenance period. Second, analysis of the inter-electrode phase synchrony and the directional information flow showed task-induced interactions in the theta band between cortical and hippocampal sites. Third, the directional information flow was from auditory cortex to hippocampus during encoding, and during maintenance, the reverse flow occurred from hippocampus to auditory cortex. This pattern was found only to the left cortical hemisphere, as expected for a language-related task. Fourth, the comparison between correct and incorrect trials suggests that the participants relied on timely information flow to produce a correct response. Our data suggests a surprisingly simple model of information flow within a network that involves sensory cortices and hippocampus (*Figure 4g*): during encoding, letter strings are verbalized as subvocal speech. The incoming information flows from sensory cortex to hippocampus (bottom-up). During maintenance, participants actively recall and rehearse the subvocal speech in their phonological loop (*Baddeley, 2003*; *Christophel et al., 2017*). The GC indicates the information flow from hippocampus to cortex (top-down) as the physiological basis for the replay of the memory items, which finally guides action.

The current study is embedded in previous studies using the same or similar tasks. Persistent firing of hippocampal neurons indicated hippocampal involvement in the maintenance of memory items (*Boran et al., 2019*; *Kamiński et al., 2017*; *Kornblith et al., 2017*). An fMRI study reports salient activity in the auditory cortex during maintenance in an auditory WM task (*Kumar et al., 2016*), which indicates that sensory cortical areas are involved in the maintenance of WM items. During encoding, the activity of local assemblies was associated with gamma frequencies and local processing (*Figure 2a b c*, *Figure 3g l*) while GC inter-areal interactions took place in theta frequencies, in line with previous reports (*Solomon et al., 2017*; *von Stein and Sarnthein, 2000*). Parietal generators of theta-alpha EEG indicated involvement of parietal cortex in WM maintenance (*Michels et al., 2008*; *Tuladhar et al., 2007*; *Näpflin et al., 2008*; *Boran et al., 2019*; *Boran et al., 2020*). The hippocampo-cortical phase synchrony (PLV) was high during maintenance of the high workload trials (*Boran et al., 2019*). Building on these previous studies, the current study focused on high workload trials and extended them by the analysis of directional information flow.

In the design of the task, we aimed to separate in time the encoding of memory items from their maintenance. In the choice of the 2 s duration for the encoding period were guided by the magic number 7±2, which may correspond to 'how many items we can utter in 2 s'(*Baddeley, 2003*; *Christophel et al., 2017*). The median Cowan's K=6.1 shows that high workload trials were indeed demanding for the participants, where both encoding and maintenance may limit performance. We therefore presented the letters both as a visual and an auditory stimulus. Certainly, maintenance processes are likely to appear already during the encoding period as maintenance neurons ramp up their activity already during encoding (*Boran et al., 2019*). Furthermore, encoding may extend past the visual stimulus (t=−3 s). We therefore focused our analysis on the last 2 s of maintenance [−2 0] s. With this task design, we found patterns of GC that were clearly distinct between encoding and maintenance.

Our study capitalizes on a unique dataset. We first benefitted from direct cortical recordings that assured the neuronal origin of the signals. We then confirmed the GC results by using the wide spatial coverage of scalp EEG, where we used beamforming to localize the cortical sources that generate the scalp EEG. The interaction between recordings from different brain regions has to be discussed with respect to volume conduction (*Trongnetrpunya et al., 2015*). On the recording level, the choice of two separate references for LFP and ECoG has been shown to avoid spurious effects in GC (*Bastos and Schoffelen, 2015*). On the level of scalp EEG analysis, we used beamforming as a source reconstruction technique (*Popov et al., 2018*) to characterize the primary neuronal generators that were localized specifically in left auditory cortex. A similar GC pattern emerged when using the signals from left temporal scalp electrodes but it was eliminated when using a Laplacian derivation. When looking at the GC frequency spectra, there was a strong frequency dependence of GC from hippocampus to ECoG (*Figure 2h*, *Figure 3d i*). Likewise, GC to EEG sources showed a strong frequency dependence (*Figure 4a*). This speaks against volume conduction because the transfer of signal through tissue by volume conduction is independent of frequency in the range of interest here (*Miceli et al., 2017*). Finally, there was a strong task dependence of GC (*Figure 2h*, *Figure 3d i*, *Figure 4a d*), again speaking against a strong contribution of volume conduction.

In the literature, there are several studies investigating the WM network. However, only few report directional interactions. One of these (*Johnson et al., 2018a*) reports cross-spectral directionality between intracranial recordings in frontal cortex and the medial temporal lobe in theta frequencies. One study on episodic memory suggests directional information flow to and from hippocampus (*Griffiths et al., 2019*). Within hippocampus, directional information flow from posterior to anterior hippocampus indicated successful WM maintenance (*Li et al., 2022*). The frequencies of GC found in the current study were in the ([4 8] Hz) theta band, in line with scalp EEG findings during WM tasks (*Sarnthein et al., 1998*; *Polanía et al., 2012*) and other tasks (*Solomon et al., 2017*) that activate oscillations in long-range recurrent connections (*Fries, 2015*; *Pesaran et al., 2018*).

Taken together, our results corroborated earlier findings on the WM network and extended them by providing a physiological mechanism for the active replay of memory items.

## Materials and methods
### Task

We used a modified Sternberg task in which the encoding of memory items and their maintenance was temporally separated (*Figure 1a*). Each trial started with a fixation period ([−6, −5] s), followed by the stimulus ([−5, −3] s). The stimulus consisted of a set of eight consonants at the center of the screen. The middle four, six, or eight letters were the memory items, which determined the set size for the trial (4, 6, or 8 respectively). The outer positions were filled with 'X', which was never a memory item. The participants read the letters and heard them spoken at the same time. After the stimulus, the letters disappeared from the screen, and the maintenance interval started ([−3, 0] s). Since the auditory encoding may have extended beyond the 2 s period, we restrict our analysis to the last 2 s of the maintenance period ([−2, 0] s). A fixation square was shown throughout fixation, encoding, and maintenance. After maintenance, a probe was presented. The participants responded with a button press to indicate whether the probe was part of the stimulus. The participants were instructed to respond as rapidly as possible without making errors. After the response, the probe was turned off, and the participants received acoustic feedback regarding whether the response was correct or incorrect. The

participants performed sessions of 50 trials in total, which lasted approximately 10 min each. Trials with different set sizes were presented in a random order, with the single exception that a trial with an incorrect response was always followed by a trial with a set size of 4. The task can be downloaded at http://www.neurobs.com/ex_files/expt_view?id=266.

## Participants

The participants in the study were patients with drug resistant focal epilepsy. To investigate a potential surgical treatment of epilepsy, the patients were implanted with intracranial electrodes. Electrodes were placed according to the findings of the non-invasive presurgical evaluation, where the epileptologists hypothesized the epileptic foci to be localized (*Zijlmans et al., 2019*). Since the presumed epileptic foci included the hippocampus in all patients, electrodes were placed in the hippocampus. In four patients, additional electrodes were placed on the cortex because an epileptic focus in the cerebral cortex was considered. The participants provided written informed consent for the study, which was approved by the institutional ethics review board (PB 2016–02055). The participants were right handed and had normal or corrected-to-normal vision. For nine participants (5–14), the PSD and PLV have been reported in an earlier study (*Boran et al., 2019*).

## Electrodes for LFP, ECoG, and EEG

The depth electrodes (1.3 mm diameter, eight contacts of 1.6 mm length, spacing between contact centers 5 mm, Ad-Tech, adtechmedical.com) were stereotactically implanted into the hippocampus for LFP recording. Subdural grid and strip electrodes (platinum electrode contacts with 4 mm$^2$ contact surface and 1 cm inter-contact distance, Ad-Tech) were placed directly on the cortex for ECoG recordings. For scalp EEG recording, cup electrodes (Ag/AgCl) were placed on the scalp and filled with electrolyte gel (Signagel, Parker Laboratories) to obtain an impedance <5 kΩ.

## Electrode localization

The stereotactic depth electrodes were localized using post-implantation CT and post-implantation structural T1-weighted MRI scans. The CT scan was registered to the post-implantation scan as implemented in FieldTrip (*Stolk et al., 2018*). A fused image of CT and MRI scans was produced and the electrode contacts were marked visually. The position of the most distal hippocampal contact was projected in a hippocampal surface (*Figure 1d*, Figure S1).

To localize the ECoG grids and strips, we used the participants' postoperative MRI, aligned to CT and produced a 3D reconstruction of the participants' pial brain surface. Grid and strip electrode coordinates were projected on the pial surface as described in *Groppe et al., 2017*; *Figure 2a* and *Figure 3a and f*.

The scalp EEG electrodes were placed at the sites of the 10–20 system by experienced technicians and no further localization was performed. While the 10–20 standard is 21 scalp electrodes, in some patients some electrode sites stayed vacant to assure the sterility of the leads to the intracranial electrodes, resulting in a median of 17 scalp sites per patient.

Some of the intracranial electrode contacts were found in tissue that was deemed to be epileptogenic and that was later resected. Still, neurons in this tissue have been found to participate in task performance in an earlier study (*Boran et al., 2019*).

## Recording setup, re-referencing, and preprocessing

All recordings (LFP, ECoG, and scalp EEG) were performed with the Neuralynx ATLAS system (sampling rate 4000 Hz, 0.5 1000 Hz passband, Neuralynx, neuralynx.com). ECoG and LFP were recorded against a common intracranial reference. Signals were analyzed in MATLAB (Mathworks, Natick MA, USA). We re-referenced the hippocampal LFP against the signal of a depth electrode contact in white matter. We re-referenced the cortical ECoG against a different depth electrode contact. The choice of two separate references for LFP and ECoG has been shown to avoid spurious functional connectivity estimates (*Bastos and Schoffelen, 2015*). The scalp EEG was recorded against an electrode near the vertex and was then re-referenced to the averaged mastoid channels. All signals were downsampled to 500 Hz. All recordings were done at least 6 hr from a seizure. Trials with large unitary artifacts in the scalp EEG were rejected. We focused on the trials with high workload (set sizes 6 and 8) for further analysis. We used the FieldTrip toolbox for data processing and analysis (*Oostenveld et al., 2011*).

## Power spectral density

We first calculated the relative PSD in the time-frequency domain (*Figure 2b*). Time-frequency maps for all trials were averaged. We used 3 multitapers with a window width of 10 cycles per frequency point, smoothed with 0.2×frequency. We computed power in the frequency range [4 100] Hz with a time resolution of 0.1 s. The PSD during fixation ([−6 −5] s) served as a baseline for the baseline correction (PSD[t] − PSD[fixation])/ PSD(fixation) for each time-frequency point.

## Phase-locking value

To evaluate the functional connectivity of hippocampus and cortex, we calculated the PLV between hippocampal LFP channels and ECoG grid (multitaper frequency transformation with two tapers based on Fourier transform, frequency range [4 100] Hz with frequency resolution of 1 Hz).

$$PLV_{i,j}\left(f\right) = \frac{1}{N}\left|\sum_{n=1}^{N}\frac{X_i(f)\cdot(X_j(f))*}{|X_i(f)|\cdot|X_j(f)|}\right|$$

where PLVi,j is the PLV between channels i and j, N is the number of trials, X(f) is the Fourier transform of x(t), and (·)* represents the complex conjugate.

Using the spectra of the 2-s epochs, phase differences were calculated for each electrode pair (i,j) to quantify the inter-electrode phase coupling. The phase difference between the two signals indexes the coherence between each electrode pair and is expressed as the PLV. The PLV ranges between 0 and 1, with values approaching 1 if the two signals show a constant phase relationship over all trials.

In our description of EEG frequency bands, we used theta [4 8] Hz, alpha [8 12] Hz, beta [12 24] Hz, and gamma >40 Hz, while the exact frequencies may differ in individual participants.

## Source reconstruction of the EEG sources

We reconstructed the scalp EEG sources using linearly constrained minimum variance (LCMV) beam-formers in the time domain. To solve the forward problem, we used a precomputed head model template and aligned the EEG electrodes of each participant to the scalp compartment of the model via interactive scaling, translation, and rotation (*ft_electrode_realign.m*). We then computed the source grid model and the leadfield matrix, wherein we determined the grid locations according to the brain parcels of the automated anatomical atlas (AAL) (*Tzourio-Mazoyer et al., 2002*). We solved the inverse problem by scanning the grid locations using the LCMV filters separately for encoding and maintenance. The EEG sources were baselined with respect to the fixation period and presented as a percent of change from the pre-stimulus baseline. We defined cortical areas from multiple parcels since AAL is a parcellation based on sulci and gyri. We performed all the steps of the source recon-struction with FieldTrip (*Oostenveld et al., 2011*) and projected the sources onto an inflated brain surface.

## Spectral Granger causality

In order to evaluate the direction of information flow between the hippocampus and the cortex, we calculated spectral non-parametric GC as a measure of directed functional connectivity analysis (*Oost-enveld et al., 2011*). We evaluated the direction of information flow in the (*Sarnthein et al., 1998*; *Li et al., 2022*; [4 20]) Hz frequency range. To compute the GC, we first downsampled the signals to the Nyquist frequency=40 Hz. We then computed the GC between hippocampal contacts and ECoG grid contacts. We also computed GC between the same hippocampal contacts and EEG sources located over the regions of interest. GC examines if the activity on one channel can forecast activity in the target channel. In the spectral domain, GC measures the fraction of the total power that is contributed by the source to the target. We transformed signals to the frequency domain using the multitaper frequency transformation method (two Hann tapers, frequency range [4 20] Hz with 20 s padding) to reduce spectral leakage and control the frequency smoothing.

We used a non-parametric spectral approach to measure the interaction in the channel pairs at a given interval time (*Bastos and Schoffelen, 2015*). In this approach, the spectral transfer matrix is obtained from the Fourier transform of the data. We used the FieldTrip toolbox to factorize the transfer function H(f) and the noise covariance matrix Σ. The transfer function and the noise cova-riance matrix were then employed to calculate the total and the intrinsic power, S(f)=H(f) Σ H×(f),

through which we calculated the Granger interaction in terms of power fractions contributed from the source to the target.

$$GC_{Y \to X} \to = ln \frac{S_{xx}(f)}{\tilde{S}_{xx}(f)}$$

where $S_{xx}(f)$ is the total power and $\tilde{S}_{xx}(f)$ is the instantaneous power. To average over the group of participants, we calculated the Granger spectra for the selected channel pairs and averaged these spectra over participants (*Figure 4a*).

To illustrate the time course of GC over time, we calculated time-frequency maps with the multi-taper convolution method of Fieldtrip (*Oostenveld et al., 2011*).

## Statistics

To analyze statistical significance, we used cluster-based non-parametric permutation tests. To assess the significance of the difference of the Granger between different directions, we compared the difference of the true values to a null distribution of differences. We recomputed GC after switching directions randomly across trials, while keeping the trial numbers for both channels constant. Then we computed the difference of GC for the two conditions. We repeated this n=200 times to create a null distribution of differences. The null distribution was exploited to calculate the percentile threshold p=0.05. In this way, we compare the difference of the dark and light spectra against a null distribution of differences. We mark the frequency range of significant GC with a blue bar for encoding (dark blue spectrum exceeds light blue spectrum, information flow from cortex to hippocampus) and with a red bar for maintenance (dark red spectrum exceeds light red spectrum, information flow from hippocampus to cortex).

To test the statistical significance of the spatial spread of contacts with high PSD, PLV, or ΔGranger, we calculated the spatial collinearity on the grid contacts against a null distribution. First, we transform the activation on grid contacts into a grid vector. We then performed 200 iterations of random trial permutations. For each iteration, we selected two subsets (50%) of trials and we calculated the scalar product between the vectors corresponding to the two subsets. The null distribution was created by randomly mixing trials from the two task periods fixation and encoding. We finally tested the statistical significance of the scalar product. The true distribution was established to be statistically distinct from the null distribution if it exceeded the 95th percentile of the null distribution.

We assess if the reconstructed EEG sources during encoding and maintenance are significantly different from the pre-stimulus baseline (fixation). We use the FieldTrip's method ft_sourcestatistics (*Oostenveld et al., 2011*), wherein we apply a non-parametric permutation approach to quantify the spatial activation pattern during the encoding of the memory items and their active replay.

Due to high average performance of the participants (91%) the number of correct and incorrect trials is imbalanced. To balance the number of correct trials with the number of incorrect trials, we randomly selected 10% of the correct trials and recomputed the GC spectra and then the net information flow (ΔGranger). We repeated this n=200 times and presented the mean ΔGranger for each participant.

For comparisons between two groups, we used the non-parametric paired cluster-based permutation test. We created a null distribution by performing N=200 random permutations.

To test the directionality of the information flow in the group of the participants, we used the group cluster-based permutation t-test from the FieldTrip toolbox (*Oostenveld et al., 2011*) with multiple comparison correction using the false discovery rate approach. Statistical significance was established at p<0.05.

## Acknowledgements

We thank the physicians and the staff at Schweizerische Epilepsie-Klinik for their assistance and the patients for their participation. We acknowledge grants awarded by the Swiss National Science Foundation (funded by SNSF 204651 to JS) and SNSF Ambizione fellowship (PZ00P3_167836 to PM) and a scholarship by Alexander S Onassis Foundation (to VD). The funders had no role in the design or analysis of the study.

## Additional information

### Funding

| Funder | Grant reference number | Author |
|---|---|---|
| Schweizerischer Nationalfonds zur Förderung der Wissenschaftlichen Forschung | funded by SNSF 204651 | Johannes Sarnthein |
| Schweizerischer Nationalfonds zur Förderung der Wissenschaftlichen Forschung | PZ00P3_167836 | Pierre Mégevand |
| Alexander S. Onassis Public Benefit Foundation | - | Vasileios Dimakopoulos |

The funders had no role in study design, data collection and interpretation, or the decision to submit the work for publication.

### Author contributions

Vasileios Dimakopoulos, Investigation, Visualization, Methodology, Writing – original draft, Writing – review and editing; Pierre Mégevand, Visualization; Lennart H Stieglitz, Lukas Imbach, Investigation; Johannes Sarnthein, Conceptualization, Supervision, Investigation, Methodology, Writing – original draft, Writing – review and editing

### Author ORCIDs

Vasileios Dimakopoulos  http://orcid.org/0000-0001-9490-565X
Pierre Mégevand  http://orcid.org/0000-0002-0427-547X
Johannes Sarnthein  http://orcid.org/0000-0001-9141-381X

### Ethics

Human subjects: The participants provided written informed consent for the study, which was approved upfront by the institutional ethics review board (PB 2016-02055).

### Decision letter and Author response

Decision letter https://doi.org/10.7554/eLife.78677.sa1
Author response https://doi.org/10.7554/eLife.78677.sa2

## Additional files

### Supplementary files
• MDAR checklist

• Supplementary file 1. ECoG and LFP recording locations. For each participant, we list the coordinates of the tip of the hippocampal electrode for LFP recording. In addition, in Participants 1–4, grid or strip electrodes were placed for ECoG recording. LFP: local field potential; ECoG: electrocorticography.

### Data availability

All data needed to evaluate the conclusions in the paper are present in the paper. The codes used to produce the results in the paper are freely available at the repository https://github.com/vdimak-opoulos/verbal_working_memory. The task can be downloaded at http://www.neurobs.com/ex_files/expt_view?id=266. Part of the data has been published earlier [36]. Additional data and code are indexed in http://www.hfozuri.ch/.

The following previously published dataset was used:

| Author(s) | Year | Dataset title | Dataset URL | Database and Identifier |
|---|---|---|---|---|
| Boran E, Fedele T, Steiner A, Hilfiker P, Stieglitz L, Grunwald T, Sarnthein J | 2020 | Dataset of human medial temporal lobe neurons, scalp and intracranial EEG during a verbal working memory task | https://www.nature.com/articles/s41597-020-0364-3 | Scientific Data, 10.1038/s41597-020-0364-3 |

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
