## [Editor Report]

The work provides important information about the communication between the auditory cortex and hippocampus during phonological working memory. The results provide crucial insights into the networks involved in this fundamental process. The results are expected to be of broad interest to readers in the fields of working memory and cognitive neuroscience in general.

---

## [Decision Letter]

**Decision letter after peer review:**

Thank you for submitting your article "Information flows from hippocampus to cortex during replay of verbal working memory items" for consideration by *eLife*. Your article has been reviewed by 3 peer reviewers, and the evaluation has been overseen by a Reviewing Editor and Laura Colgin as the Senior Editor. The following individual involved in review of your submission has agreed to reveal their identity: Joel Berger (Reviewer #1).

Essential revisions:

The referees all found the work interesting but raise interpretational issues in their detailed comment that all need to be addressed. We have highlighted a number of these below:

1) The data need to be reported less selectively to give readers an idea of the consistency of the effects emphasised.

2) One referee raises the issue of possible volume conduction which can be addressed with re-referencing of electrodes.

3) Many claims are based on descriptive data as opposed to quantification of the effect claimed and there are statistical issues raised including correction for multiple comparisons.

4) The authors need to comment on a more systematic examination of effects in different frequency ranges including the γ range.

If these issues can be addressed the suggested model will be of broad interest, but an extensive revision will be required.

*Reviewer #1:*

The authors have shown a unique set of recordings, wherein they have collected intracranial data from parietal cortex and hippocampus, as well as scalp EEG in a number of subjects. With this unique advantage, they have examined directionality of connectivity between various regions during a working memory task. Given the growing evidence for the role of hippocampus in working memory, understanding its connectivity to the rest of the brain provides a crucial insight into the network involved in such a fundamental process. Whilst the existing content is generally of a high standard, and the analyses seem sound, there are some areas of considerable brevity that would benefit from expansion. Below are my comments on the manuscript.

Discussion: This is surprisingly short Discussion section. I feel this should be expanded considerably, such as including some of the information that I have discussed below regarding potential considerations of the task (e.g bimodal nature), discussion of the PLV results in the context of previous non-directional findings, the differences observed between correct and incorrect trials, considering in more detail the other behavioral consequences of these results. These suggestions are not necessarily exhaustive but are all points I believe should be included.

Comments on Results section in general:

– All the results in this section seem to refer to a single electrode for each subject. It would be beneficial to know whether these electrodes were representative of activity from surrounding electrodes or not. That is, how generalizable are the PSD results shown here.

– Also, many of these results are very descriptive. Whilst in some specific scenarios this is unavoidable, for the purposes of reporting results from PSD (for example) it is definitely possible to report details such as the degree of power increase. At present, this reads more like a Discussion section that an informative Results section.

– It would be helpful to see an overlay of the parietal electrode with a topographic map of the scalp EEG recording, to truly appreciate the spatial overlap between the electrode and the generator.

Figures in general: many of the figures appear to refer only to single subjects. It would be useful to have more detailed summary information across subjects to understand how reliable/variable these effects are.

Data availability section: The bit on previously published datasets confused me a little. Is this published dataset included as part of this article? It isn't so clear in the manuscript whether these are previously published data. If they are, this should be made more apparent.

Line 29: Phrasing – I would add the words "rather than sequentially" here to help readers with interpretation of why this separates out encoding from maintenance

Line 64: This can actually extend as low as δ band (see Leszczynski et al., 2015, Cell Reports; Kumar et al., 2021, Neuropsychologia).

Line 88: Do the authors have behavioral data or prior knowledge of how long it takes (on average) to encode 4, 6 or 8 letters? That is, how much of the 'encoding' period is truly encoding, rather than an initial encoding followed by maintenance. Or in a similar manner, how much of maintenance is still residual encoding.

Line 90: Was there a particular reason as to why the encoding phase was bimodal? Do the authors think this may have influenced their results?

Line 94-95: Was this an instruction to the participants? If so, I would put this more explicitly, i.e. "participants were instructed to rehearse…". Of course, one cannot know for certain whether individual subjects employed this strategy.

Figure 2f: Where is this change in Granger relative to? A particular baseline window.

Line 293: Were any electrodes here included in a seizure foci? Was anything done to ensure that seizure activity did not affect recordings (e.g. not recording within xxx hours of a seizure)?

Line 301: Was anything done to deal with artefacts on the ECoG/sEEG electrodes? I.e. were trials with unusually large amplitudes, potentially indicative of muscle artefact (a known contaminant) removed?

Line 325-326: I am confused by this. You say that the individual frequencies may differ between participants – do you mean in terms of the peak frequency, or were different bands used for each subject? If different bands, why?

All power spectral density plots: I assume these are relative to baseline. Are they statistically-thresholded in any way?

*Reviewer #2:*

Dimakopoulos et al. use intracranial data in humans to ask whether information flow is primarily cortical to hippocampal or the reverse during the encoding and retrieval stages of a working memory task. They find a highly reliable pattern where information in the α/β range flows from auditory cortex to hippocampus during encoding and in the reverse direction during maintenance of items in WM. The authors show this pattern in a sub-selection of ECoG recordings and go on to show it is present in virtually all subjects at the EEG to intracranial hippocampus level. In addition, this directional pattern breaks down during incorrect trials. However, the current analysis suffers from possible contamination by volume conduction.

The study is unique in its data set and provides a valuable look into hippocampal cortical interactions during WM. However, there are multiple technical questions remaining. One of the limitations is that the study investigated primarily interactions in the α/β range when looking at interactions. In contrast, their power spectral results show increases in γ during encoding, and other studies have emphasized a role for γ in feedforward routing. Did the authors perform a granger causality analysis in γ?

1. The reported PLV values (e.g., in figure 2d) of 0.3 – 0.6 are quite high. This level of PLV is usually observed when there is some amount of volume conduction present. Could the authors repeat the PLV and GC analysis using a local bipolar re-referenced signal? For more information on why this is a problem please see Trongnetrpunya, Nandi, Ding, et al., Frontiers in Systems Neuroscience.

2. Did the authors find any detectable GC or PLV in the γ frequency range, particularly during the encoding epoch?

3. Across subjects, the authors report that the Sensory to Hippocampus information flow is dominant during sensory processing and the reverse during delay processing (Figure 5). In Figure 5b, it is impressive how the effect holds in each and every individual subject. My question is how were the data selected in figure 5b? Is that an average of all GC pairs linking all available hippocampus leads to / from all available EEG leads, or was there a data selection that occurred? What would the results look like if only the EEG leads are selected that overly to auditory/temporal cortex? Is the topography the same or different when performing the analysis on e.g., frontal / occipital / parietal EEG leads? In other words, how spatially specific is the effect?

*Reviewer #3:*

Dimakopoulos and colleagues investigate connectivity and flow of information during encoding and maintenance of Working Memory. They use unique data, which combine human intracranial recordings from depth electrodes with ECOG and EEG. This data, combined with Granger causality analysis (GC), provides interesting results that signal from cortex (mostly from EEG electrodes located over temporal cortex) is flowing to hippocampus during encoding and this flow is reversed during maintenance. Authors interpret this as a sign of bottom-up and top-down processing. I believe that chosen methods for signal analysis are appropriate.

However, paper contains several statements that are unsupported by statistics and there is no clear information about why some decisions in the analysis process were made. This could give an impression that the analysis is built from arbitrarily chosen single case examples. I believe that because of below listed flaws results of the analysis do not support conclusions.

1) Authors do not use correction for multiple comparisons – this cast doubts on the strength of obtained results.

2) There is no criterion given for ECOG electrodes selected to the analysis.

For instance, authors state that for participant 1 for C2 electrode, increased γ power during encoding proves that this electrode was over auditory cortex but there is no systematic analysis of γ power. From the results we can observe that this electrode has the strongest GC with hippocampus what suggests that it was used because of this characteristic what looks like double dipping.

3) Why frequencies observed in PLV and GC are so different? For instance, in supplementary Figure 1 PLV shows significant differences in 18-30 Hz but GC is calculated for 8-18 Hz. Such large differences in frequencies suggest some inconsistencies in the analysis.

4) For analyses depicted in Figure 4 and 5 it is unknown how the highest GC is defined (is it a mean from all frequencies?) Furthermore, there is no systematic measurement or criterion that would support that indeed chosen electrodes have the highest GC.

5) All analysis conducted in the time domain (time to frequency and GC) does not contain any statistics supporting validity of the proposed conclusions.

6) There is no data that supports statement that patients used verbalization. Although material is verbal authors cannot rule out that subject uses different modalities to support information maintenance.

7) In figure 3i significance for encoding (blue) is marked where there are no differences between dark blue and light blue conditions. This suggests that the significance is computed not between conditions but versus null distribution. In general, in the paper, it is unclear if comparison using null distribution or between conditions is being calculated.

8) In Figure 2b – why EEG, and not ECOG, signal is being analyzed?

9) I appreciate that authors present single subject results in Figure 4 but I believe group level analysis should be also performed to support conclusions.

10) Does the quality/magnitude of GC estimation depends on the number of trials? If so

for Figure 5b authors show subsample number of trials in the correct condition to match the number of trials in the incorrect condition.

11) Authors should state how many sessions were used in each comparison.

12) In the statement:

"The GC was lower than for ECoG, as expected for the lower signal amplitude of scalp EEG." I do not know what "lower" means – is it a magnitude or frequency of GC? If frequency, how does it relate to the scalp EEG amplitude?

13) Throughout the paper it seems that "ΔGranger" is used to describe a different thing.

[Editors’ note: the authors submitted for reconsideration following the decision after peer review. What follows is the decision letter after the second round of review.]

Thank you for resubmitting the paper entitled "Information flows from hippocampus to cortex during replay of verbal working memory items" for further consideration by *eLife*. Your revised article has been evaluated by a Senior Editor and a Reviewing Editor. The revision has also been re-evaluated by three referees. We apologise for the delay in this decision over the holiday period. We are sorry to say that we have decided that this submission will not be considered further for publication by *eLife*.

We found the approach to working memory based on effective connectivity between sensory cortex and hippocampus during working memory maintenance and retrieval intriguing and novel. All of the referees appreciated the efforts that have been made to improve the manuscript. The principal reason for rejection is the statistical interpretation of the data raised by all three referees, who are concerned whether the analyses allow robust inference about effects in the population. Referee 1 highlights differences between subjects in the frequency bands of effective connectivity. The effects are not robust to re-referencing recommended by referee 2, who does not consider a common reference adequate for this work. Referee 3 is concerned about multiple comparisons.

*Reviewer #1:*

I appreciate the efforts the authors have taken to improve the manuscript. The additional information does aid interpretation of the study. I still have a couple of remaining core issues to be sorted.

1) In the abstract, it states that information flow from auditory cortex to hippocampus peaked in the range of 8-18 Hz. In the newly inserted (and very helpful) table it seems that the range is anywhere from 4 to 21 Hz. Unless I am missing something, it appears that the authors are still making conclusions based on only Participant 1 (who showed significance in the 8-18 Hz range). I feel that this confusion runs throughout the manuscript and should be corrected – that is, any points where generalizations are made from a single subject that are not truly consistent across subjects should be corrected and the variability should be highlighted/discussed.

2) Similarly, the frequency band of hippocampus > auditory cortex is often not congruent with the AC > hippocampus within subjects, so it is too simplistic to say this was reversed during maintenance. Moreover, this difference in frequencies within subjects should be discussed.

*Reviewer #2:*

The authors have attempted to answer my concerns by performing a bipolar and Laplacian analysis to the data prior to the connectivity analysis. Unfortunately, this revealed that effects were not preserved after applying local re-referencing. I appreciate the authors' arguments that some effects have a large spatial structure that is removed by bipolar. However, the electrodes used by the authors are quite far apart and a priori I would think that a referencing across a large spatial structure should preserve true effects while removing spurious ones.

Unfortunately, given simulation work as in the article I referred to in my original review (Trongnetrpunya, Nandi, Ding, et al., Frontiers in Systems Neuroscience), the false positive rate is unacceptably high when Granger and related methods are applied to data sharing a common reference. The current results therefore may be incorrect and cannot be interpreted in their present form.

The authors also apparently have cherry picked their data in Figures 4 and 5 by first taking channels that show a change in Granger, and then analyzing those changes across participants. It is like forcing the data to behave as we would like, another example of a flawed approach taken by the present manuscript. Given these technical/methodological issues, I cannot recommend the manuscript for publication in the present state.

I do want to commend the authors for recording this dataset, and performing directed connectivity analysis, which is rare for these types of data. I would encourage the authors to think more deeply about how to assess directed functional connectivity robustly and reformulate their work with this in mind.

*Reviewer #3:*

The authors have revised their manuscript and attempted to address all of the comments raised by myself and the other reviewers.

I have few remaining issues.

If authors compute the difference against null distribution not between conditions (for example light blue and dark blue) in Figure 3d,i and in Figure 4 – How statement "dark blue spectrum exceeds light blue spectrum, information flow from cortex to hippocampus" is correct? It seems that, for each light colour result (hippocampus to cortex during encoding, cortex to hippocampus during maintenance) there should be a result of a statistical test and it seems none of these tests were performed. The caption of Figure 4 where authors use ΔGranger seems also incorrect as authors compute statistics from null distribution.

Unfortunately, I believe that issue with multiple comparisons was not addressed entirely. For instance, the analysis in the frequency domain does not use any correction for multiple comparisons. Because of that it is unknown if small differences observed in results are not effects of multiple comparisons.

Additionally for Figure 5 b and c analysis it seems that the comparison is not designed correctly. For figure 5b the data is picked using significant reversal of net GC (ΔGranger) – this is not done for Figure 5c analysis so it is hard to draw any conclusions from this contrast.

About the group statistics issue. Figure 5a although presents average results does not show any statistics thus it is unknown if any of those information flows is significant. Figure 4 can give me an intuition that the result is present, but I have no idea about how significant it is on the group level. I believe that paper would benefit from the group statistics.

[Editors’ note: further revisions were suggested prior to acceptance, as described below.]

Thank you for resubmitting your work entitled "Information flows from hippocampus to auditory cortex during replay of verbal working memory items" for further consideration by *eLife*. Your revised article has been evaluated by Laura Colgin (Senior Editor), a Reviewing Editor, and the original reviewers.

The work provides new insights into the communication between sensory cortex and hippocampus during working memory.

The revised work has been seen by all of the original referees who agree it is substantially improved. They have raised a number of further points about analysis and exposition we would like you to consider when preparing the final article for acceptance.

1. No details are provided for the scalp EEG system. How many electrodes? Was this an active system? Filtering at all? Many of these points are particularly relevant and important for interpreting the beamforming data. It says that the electrodes were aligned for each participant – how was electrode localization performed? Using what system – Polhemus?

2. Was all source reconstruction done with a template MRI? If so, wouldn't it have been appropriate to use each individual's MRI for better localisation, given that you surely had these available? You could then do the group averages, but at least the individual subject data would be more accurately localised.

3. The Granger causality analyses are reported for only three of the subjects between hippocampal LFP and ECoG data. Did the other 12 subjects not have any cortical electrodes then, even without parietal coverage, only hippocampal LFPs? This isn't at all clear from the descriptions in the manuscript and seems surprising that data would only be available from hippocampus, and no other temporal regions, even if parietal coverage was lacking.

4. It would be beneficial to include a display of all the contact locations for hippocampus, for transparency and interpretability purposes, along with (or perhaps instead) a supplement including all MNI coordinates of localised electrodes.

5. L110-110: It would be more consistent with other literature to describe this frequency range as "high α/low β".

6. At a number of points, the authors state that during maintenance, rehearsal is performed as a melody. Where does this assumption come from? That is, nowhere is it clear that these letters are transformed into a melodic representation (i.e. related to a musical sequence). This seems overly speculative.

7. The beamforming shows the area specificity for the effects in auditory cortex. The volume conduction paragraph should take this into account, but should also mention that the PLV and Laplacian based referencing scheme eliminated the effects. There should be a discussion about the possible reasons as to why this was observed in the author's data analysis.

8. On a small technical note, raw Granger should not be multiplied by 100%. Please revert back to the raw Granger values, as this would make the work more comparable to other studies, and these numerical units are meaningful. It is OK to express GC percentage differences in the cortex to hippocampus vs. hippocampus to cortex directions during the task modulation, however, to emphasize the changing direction patterns.

9. Relative power plots are not showing any decreases in power relative to fixation. How were the data baselined? They vary from 0 to 1. What about power decreases from fixation?

10. Lines 200-206 when reporting the change in Granger using a permutation test, but no p-values are given.

*Reviewer #2:*

I appreciate some of the additions that the authors have included. I do have further suggestions on the manuscript.

No details are provided for the scalp EEG system. How many electrodes? Was this an active system? Filtering at all? Many of these points are particularly relevant and important for interpreting the beamforming data. It says that the electrodes were aligned for each participant – how was electrode localization performed? Using what system – Polhemus?

Was all source reconstruction done with a template MRI? If so, wouldn't it have been appropriate to use each individual's MRI for better localisation, given that you surely had these available? You could then do the group averages, but at least the individual subject data would be more accurately localised.

The granger causality analyses are reported for only three of the subjects between hippocampal LFP and ECoG data. Did the other 12 subjects not have any cortical electrodes then, even without parietal coverage, only hippocampal LFPs? This isn't at all clear from the descriptions in the manuscript and seems surprising that data would only be available from hippocampus, and no other temporal regions, even if parietal coverage was lacking.

It would be beneficial to include a display of all the contact locations for hippocampus, for transparency and interpretability purposes, along with (or perhaps instead) a supplement including all MNI coordinates of localised electrodes.

*Reviewer #3:*

The authors have quantified patterns of information flow during a working memory task between cortex and hippocampus. They capitalize on a unique dataset where human intracranial data have been collected from the hippocampus, in parallel with human electrocorticography and/or electroencephalography. The authors use a statistical method to infer functional directed connectivity. This reveals information flowing from cortex to hippocampus during encoding but the reverse direction during maintenance. This study is quite unique in its combination of methodology with data that can precisely track the anatomical sources of neural computations during a memory task.

Recommendations for the authors:

In particular, the nice use of beamforming show the area specificity for the effects in auditory cortex. The volume conduction paragraph should take this into account, but should also mention that the PLV and Laplacian based referencing scheme eliminated the effects. There should be a discussion about the possible reasons as to why this was observed in the author's data analysis.

On a small technical note, raw Granger should not be multiplied by 100%. Please revert back to the raw Granger values, as this would make the work more comparable to other studies, and these numerical units are meaningful. It is OK to express GC percentage differences in the cortex to hippocampus vs. hippocampus to cortex directions during the task modulation, however, to emphasize the changing direction patterns.

Relative power plots are not showing any decreases in power relative to fixation. How were the data baselined? They vary from 0 to 1. What about power decreases from fixation?

Lines 200-206 when reporting the change in Granger using a permutation test, but no p-values are given.

---

## [Author Response]

Reviewer #1:The authors have shown a unique set of recordings, wherein they have collected intracranial data from parietal cortex and hippocampus, as well as scalp EEG in a number of subjects. With this unique advantage, they have examined directionality of connectivity between various regions during a working memory task. Given the growing evidence for the role of hippocampus in working memory, understanding its connectivity to the rest of the brain provides a crucial insight into the network involved in such a fundamental process. Whilst the existing content is generally of a high standard, and the analyses seem sound, there are some areas of considerable brevity that would benefit from expansion. Below are my comments on the manuscript.Discussion: This is surprisingly short Discussion section. I feel this should be expanded considerably, such as including some of the information that I have discussed below regarding potential considerations of the task (e.g bimodal nature), discussion of the PLV results in the context of previous non-directional findings, the differences observed between correct and incorrect trials, considering in more detail the other behavioral consequences of these results. These suggestions are not necessarily exhaustive but are all points I believe should be included.

We found muscle artefact in the scalp EEG but not in the ECoG/sEEG. Trials with muscle artefact in the scalp EEG were removed.

Following the Reviewer’s comment, we have now mentioned this in more detail in the Methods (Line 419). Furthermore, we have created a new Table 1 where we list the number of correct trials analyzed for each participant.

Comments on Results section in general:– All the results in this section seem to refer to a single electrode for each subject. It would be beneficial to know whether these electrodes were representative of activity from surrounding electrodes or not. That is, how generalizable are the PSD results shown here.

Participant 1 had a large grid over posterior cortex and we now present several topographical plots. We were indeed able to see some similarities between adjacent electrodes.

Following the Reviewer’s comment, we have now moved the topographical plots of GC from the supplementary material to the new Figure 2. Furthermore, we have added topographical plots for PSD in the new Figure 2 a, d, for PLV in Figure 2 h, and for ΔGranger in Figure 2 i, j. Regarding the GC to scalp EEG, we have created Table 2 that highlights all scalp electrode sites where ΔGranger was significant during maintenance. It turned out that ΔGranger was always > 0 (net information flow from hippocampus to cortex) and that the majority of these scalp electrodes were on central or temporal sites, suggesting involvement of auditory cortex.

– Also, many of these results are very descriptive. Whilst in some specific scenarios this is unavoidable, for the purposes of reporting results from PSD (for example) it is definitely possible to report details such as the degree of power increase. At present, this reads more like a Discussion section that an informative Results section.

We show PSD in time-frequency plots. As the reviewer rightly requests, these show the degree of power increase with respect to fixation [-6 -5] s as a baseline according to (PSD(t) – PSD(fixation))/ PSD(fixation). For example, in Figure 1b, γ PSD during encoding has increased more than twice with respect to PSD during fixation.

Following the Reviewer’s comment, we have now entered the number of maximal increases in the Results (Line 124, 126, 142).

We focus on changes in GC as the main novel message of our study. Here we perform extensive computational statistics. In earlier studies with the same task, we have extensively analyzed scalp EEG (Michels 2008), PLV between hippocampus and scalp EEG (Boran 2019), and single neuron firing in the hippocampus (Boran 2019). We validate our results by statistical testing to ascertain that the observed changes in GC were indeed associated with the task.

– It would be helpful to see an overlay of the parietal electrode with a topographic map of the scalp EEG recording, to truly appreciate the spatial overlap between the electrode and the generator.

We thank the reviewer for this suggestion.

We have now added the scalp EEG recording sites to the corresponding panels in Figure 2 (Line 524-525) and Figure 3 (Line 581-582).

Figures in general: many of the figures appear to refer only to single subjects. It would be useful to have more detailed summary information across subjects to understand how reliable/variable these effects are.

Figure 1 and Figure 5 aggregate information across subjects. The panels of Figure 2 and Figure 3 are arranged to facilitate comparison between results of participants 1,2,3. The panels of Figure 4 allow comparison across all participants.

We have created a new Table 1 and Table 2 to facilitate the understanding of how reliable the effects are across subjects.

Data availability section: The bit on previously published datasets confused me a little. Is this published dataset included as part of this article? It isn't so clear in the manuscript whether these are previously published data. If they are, this should be made more apparent.

The data of our earlier study (Boran 2019) has been published (Boran 2020). In that study we had analyzed the PLV between scalp EEG and hippocampus sEEG in 9 participants. We include these participants here. As a new analysis, we calculate GC between scalp EEG and hippocampus sEEG.

We have created a table listing the data that has been published previously and the data that is newly analyzed for this manuscript.

Line 29: Phrasing – I would add the words "rather than sequentially" here to help readers with interpretation of why this separates out encoding from maintenance

Following the Reviewer’s comment, we have now added "rather than sequentially" (Line 31).

Line 64: This can actually extend as low as δ band (see Leszczynski et al., 2015, Cell Reports; Kumar et al., 2021, Neuropsychologia).

Following the Reviewer’s comment, we have now added this information and cited (Kumar 2021), (Line 65).

Line 88: Do the authors have behavioral data or prior knowledge of how long it takes (on average) to encode 4, 6 or 8 letters? That is, how much of the 'encoding' period is truly encoding, rather than an initial encoding followed by maintenance. Or in a similar manner, how much of maintenance is still residual encoding.

In the design of the task we were guided by the magic number 7±2, which may correspond to “how many items we can utter in 2 seconds” (Baddeley 2003). Certainly, maintenance neurons ramp up their activity already during encoding (Figure 2 in (Boran 2019)). In a similar manner, encoding may extend past the visual stimulus (t = -3 s). We therefore analyze only the last two seconds of maintenance [-2 0] s. With this design, we found patterns of GC that were distinct between encoding and maintenance.

We have now added this information in the discussion (Line 285 – 296)

Line 90: Was there a particular reason as to why the encoding phase was bimodal? Do the authors think this may have influenced their results?

Several of the patients were not able to read 8 letters within the encoding period of 2 s. Reading the letters to them greatly improved their performance. At the same time, the auditory encoding may have extended beyond the 2 s period. We therefore restrict our analysis to the last 2 s of the maintenance period.

Following the Reviewer’s comment, we have now entered this in Methods (Line 362-364).

Line 94-95: Was this an instruction to the participants? If so, I would put this more explicitly, i.e. "participants were instructed to rehearse…". Of course, one cannot know for certain whether individual subjects employed this strategy.

We instructed participants to rehearse. We further asked them whether they had in fact employed this strategy. The answer was yes in all subjects.

Following the Reviewer’s comment, we have now entered the following in Results (Line 100-101).

Figure 2f: Where is this change in Granger relative to? A particular baseline window.

ΔGranger denotes the net information flow (GC_hippcortex_ – GC_cortexhipp_). A positive ΔGranger indicates predominant information flow from hippocampus to cortex (dark red spectrum exceeds light red spectrum), e.g. during maintenance (new Figure 2h). The new Figure 2l illustrates ΔGranger for each time point for contact C2.

We have now described this in more detail in Results and the figure caption (Line 170 -172, 179-180, 199, 576-578)

Line 293: Were any electrodes here included in a seizure foci? Was anything done to ensure that seizure activity did not affect recordings (e.g. not recording within xxx hours of a seizure)?

Some of the electrodes were in tissue that was deemed to be epileptogenic and that was later resected. Still, neurons in this tissue have been found to participate in task performance (Boran 2019). All recordings were done at least 6 h from a seizure.

Following the Reviewer’s comment, we have now entered this in the Methods and we now report whether the hippocampal recording site was part of the seizure onset zone in Table 1 (Line 402-404).

Line 301: Was anything done to deal with artefacts on the ECoG/sEEG electrodes? I.e. were trials with unusually large amplitudes, potentially indicative of muscle artefact (a known contaminant) removed?

We found muscle artefact in the scalp EEG but not in the EcoG/sEEG. Trials with muscle artefact in the scalp EEG were removed.

Following the Reviewer’s comment, we have now mentioned this in more detail in the Methods (Line 419). Furthermore, we have created a new Table 1 where we list the number of correct trials analyzed for each participant.

Line 325-326: I am confused by this. You say that the individual frequencies may differ between participants – do you mean in terms of the peak frequency, or were different bands used for each subject? If different bands, why?

The frequency ranges result from the statistical analysis. The frequency ranges turned out to vary across participants. The variability of frequency bands across participants has been shown earlier for this task (Boran 2019, Michels 2008) and is a general phenomenon in cognitive EEG research.

We now clarify this phrase and extend the discussion (Line 315 – 319).

All power spectral density plots: I assume these are relative to baseline. Are they statistically-thresholded in any way?

Correct, the power spectral density plots are relative to the fixation period [-6 -5] s. They are not thresholded.

We now add this information to the Figure captions (Line 569- 570, 610 – 611).

Reviewer #2:Dimakopoulos et al. use intracranial data in humans to ask whether information flow is primarily cortical to hippocampal or the reverse during the encoding and retrieval stages of a working memory task. They find a highly reliable pattern where information in the α/β range flows from auditory cortex to hippocampus during encoding and in the reverse direction during maintenance of items in WM. The authors show this pattern in a sub-selection of EcoG recordings and go on to show it is present in virtually all subjects at the EEG to intracranial hippocampus level. In addition, this directional pattern breaks down during incorrect trials. However, the current analysis suffers from possible contamination by volume conduction.The study is unique in Its data set and provides a valuable look into hippocampal cortical interactions during WM. However, there are multiple technical questions remaining. One of the limitations is that the study investigated primarily interactions in the α/β range when looking at interactions. In contrast, their power spectral results show increases in γ during encoding, and other studies have emphasized a role for γ in feedforward routing. Did the authors perform a granger causality analysis in γ?

Please find our detailed response below to Point 2.

1. The reported PLV values (e.g., in figure 2d) of 0.3 – 0.6 are quite high. This level of PLV is usually observed when there is some amount of volume conduction present. Could the authors repeat the PLV and GC analysis using a local bipolar re-referenced signal? For more information on why this is a problem please see Trongnetrpunya, Nandi, Ding, et al., Frontiers in Systems Neuroscience.

We apologize for presenting PLV in Figure 2d with a color scale only in the range [0.3 0.6]. In the new Figure 2g (Line 598), we now show the full range of PLV < 0.6. The PLV to parietal recording sites is < 0.1, i.e. with high anisotropy across contacts. Similarly, GC is highly anisotropic both for eCoG (Figure 2) and for scalp EEG (Table 2). This anisotropy speaks against a significant amount of volume conduction present.

Following the Reviewer’s suggestion, we have repeated PLV and GC analysis with a Laplacian montage for the ECoG of Participant 1 and a bipolar montage for the ECoG of all three Participants with ECoG. There was neither PLV nor GC. However, Laplacian and bipolar montages focus on highly localized neuronal assemblies and provides somewhat complementary information to the referential montage (Nunez 1995). The activity of local assemblies is rather associated with higher frequencies and local processing. Inter-areal interactions tend to take place rather in lower frequencies (Solomon 2017, von Stein 2000), which require a referential montage to be appreciable in the signal.

There was a strong frequency dependence of PLV to ECoG (Participant 1, Figure 2g) and to scalp EEG (Figure 6 in (Boran 2019)). Likewise, GC to ECoG and to scalp EEG showed a strong frequency dependence (Figures 2, 3, 4). This speaks against volume conduction because the transfer of signal through tissue by volume conduction is independent of frequency in the range of the EEG studied here (Miceli 2017, Nunez 1995).

There was a strong task dependence of both PLV (Boran 2019) and GC (Figures 2, 3, 4). This also speaks against a strong contribution of volume conduction.

In conclusion, while we cannot rule out a contribution of volume conduction in our data, it seems to be small compared to the effects in PLV and GC that were induced by the processing of the cognitive task.

We have now added a paragraph on volume conduction in the discussion (Line 297 – 313). We have now reproduced the PLV results of (Boran 2019) in a new supplementary Table S1, in which we have added the PLV results of the additional participants of this study.

2. Did the authors find any detectable GC or PLV in the γ frequency range, particularly during the encoding epoch?

We agree that there was a strong effect in γ during encoding. Therefore, we always extended our analyses to γ frequencies. Unfortunately, in our GC analysis >30 Hz we never found significant difference in directionality. Please see Author response image 1 for the analysis up to 100 Hz.

We therefore presented spectra only <30 Hz. As can be seen in all GC plots, the four GC spectra show little difference already around 30 Hz.

We now report this null-finding in Results (Line 178)

**Author response image 1. sa2fig1:** 

3. Across subjects, the authors report that the Sensory to Hippocampus information flow is dominant during sensory processing and the reverse during delay processing (Figure 5). In Figure 5b, it is impressive how the effect holds in each and every individual subject. My question is how were the data selected in figure 5b? Is that an average of all GC pairs linking all available hippocampus leads to / from all available EEG leads, or was there a data selection that occurred? What would the results look like if only the EEG leads are selected that overly to auditory/temporal cortex? Is the topography the same or different when performing the analysis on e.g., frontal / occipital / parietal EEG leads? In other words, how spatially specific is the effect?

The spatial sampling of scalp EEG electrodes is very sparse when compared with the anatomical specification of the cortex. We can not a priori assume that adjacent electrodes show similar behavior. For Figure 4 and Figure 5 we therefore selected the electrode with the highest GC value where the reversal of net GC (ΔGranger) was statistically significant. Figure 4 and Figure 5 present data from the same scalp electrodes, respectively.

We have created a new Table 2 where we list the maximal GC during maintenance in the frequency band where the reversal of net GC (ΔGranger) was statistically significant. In 14/15 participants, the maximal GC appeared in electrodes over central or temporal cortex.

We have now mentioned this finding in the Results (Line 216 – 219) and in the Discussion (Line 332 – 337).

Reviewer #3:Dimakopoulos and colleagues investigate connectivity and flow of information during encoding and maintenance of Working Memory. They use unique data, which combine human intracranial recordings from depth electrodes with ECOG and EEG. This data, combined with Granger causality analysis (GC), provides interesting results that signal from cortex (mostly from EEG electrodes located over temporal cortex) is flowing to hippocampus during encoding and this flow is reversed during maintenance. Authors interpret this as a sign of bottom-up and top-down processing. I believe that chosen methods for signal analysis are appropriate.However, paper contains several statements that are unsupported by statistics and there is no clear information about why some decisions in the analysis process were made. This could give an impression that the analysis is built from arbitrarily chosen single case examples. I believe that because of below listed flaws results of the analysis do not support conclusions.1) Authors do not use correction for multiple comparisons – this cast doubts on the strength of obtained results.

Following the Reviewer’s comment, we add data from more electrodes and test the spatial distribution of the results on the grid electrode.

We have now added further analyses and panels to Figure 2 and Figure 3 and created a new Table 2. In Figure 2, the new panels show that the grid contacts C2 and H3, which we mainly present, are surrounded by contacts with similar behavior (Figure 2 a, d, g, i, j). To provide a sound statistical basis, we have now tested the spatial distribution of PSD, PLV and GC on the grid contacts against a null distribution. The activation on grid contacts was transformed into a grid vector. The spatial collinearity of two grid vectors is captured by their scalar product. We tested the statistical significance of the spatial spread of contacts with high ΔGranger (8-14 Hz) during maintenance ([-2 0] s). We first reshaped the ΔGranger values on the grid into a vector over 200 iterations of random trial permutations. For each iteration we selected two subsets of trials and we calculated the scalar product between the two subsets.The true distribution of scalar products occurs only above the 95^th^ percentile of the null distribution and is thereby statistically significant (Figure 2 k).

For the scalp electrodes of all participants, the Table 2 shows that significant GC occurred specifically to central/temporal scalp EEG electrodes in 14/15 participants (p=0.005, two sided non-parametric permutation test).

We have now added this information to the Results (Line 186 – 197) and the Discussion (Line 327 – 334) sections.

2) There is no criterion given for ECOG electrodes selected to the analysis.For instance, authors state that for participant 1 for C2 electrode, increased γ power during encoding proves that this electrode was over auditory cortex but there is no systematic analysis of γ power. From the results we can observe that this electrode has the strongest GC with hippocampus what suggests that it was used because of this characteristic what looks like double dipping.

Following the Reviewer’s comment, we have added further analyses and panels to Figure 2. Encoding elicited γ power increase at several temporal contacts (among them C2) and occipital contacts. We tested the statistical significance of the increase for each contact. Figure 2a shows the z-score of those contacts in red where the increase reached statistical significance of the increase (z-score >1.96, permutation t-test). In a separate test (NOT double-dipping!), we identified the grid contacts with significant increase in PLV and GC in Figure 2 g, i, j. These turn out to be on temporal cortex as well.

3) Why frequencies observed in PLV and GC are so different? For instance, in supplementary Figure 1 PLV shows significant differences in 18-30 Hz but GC is calculated for 8-18 Hz. Such large differences in frequencies suggest some inconsistencies in the analysis.

In a mechanistic view of communication theory, one assumes that a node displays high PSD, high PLV, high GC, and all of this in the same time window and in the same frequency band. Our data come close to this view but the frequency bands do not match. Since this mismatch follows directly from the data, we would rather call it an inconsistency between the available analysis methods and our underlying assumptions.

Following the Reviewer’s comment, we now highlight these considerations in the Discussion (Line 315 – 323).

4) For analyses depicted in Figure 4 and 5 it is unknown how the highest GC is defined (is it a mean from all frequencies?) Furthermore, there is no systematic measurement or criterion that would support that indeed chosen electrodes have the highest GC.

The highest GC is defined as the mean in the band of significant GC change. We present here the electrode pairs with the highest ΔGranger values. It turned out that in 14/15 participants, the highest GC occurred in temporal electrodes over auditory cortex or in participants where temporal sites were not recorded from, the highest GC occurred at the neighboring electrode sites C3 or C4 (Table 2, p=0.005, two sided non-parametric permutation test).

We have now prepared a Table 2 that lists all electrodes and GC values. We now mention our criterion not only in the Methods (Line 466, 482 – 491) but also in the Results sections (Line 242 – 244).

5) All analysis conducted in the time domain (time to frequency and GC) does not contain any statistics supporting validity of the proposed conclusions.

We agree with the Reviewer that the time frequency maps are illustrative only. Statistical testing was performed for the GC frequency spectra in the time windows encoding [-5 -3] s and maintenance [-2 0] s. The time-frequency plots are labelled as mere illustrations throughout the text.

6) There is no data that supports statement that patients used verbalization. Although material is verbal authors cannot rule out that subject uses different modalities to support information maintenance.

During the maintenance period, participants rehearsed the verbal representation of the letter strings subvocally, i.e. mentally replayed the memory items. Participants had been instructed to employ this strategy and they confirmed after the sessions that they had indeed employed this strategy.

We now add this information more prominently in Results (Line 100).

7) In figure 3i significance for encoding (blue) is marked where there are no differences between dark blue and light blue conditions. This suggests that the significance is computed not between conditions but versus null distribution. In general, in the paper, it is unclear if comparison using null distribution or between conditions is being calculated.

The Reviewer is correct. We compute significance against a null distribution as stated in the Methods: “To assess the significance of GC, we compared the true values to a null distribution. We recomputed GC after shuffling the trial number for a single channel in the pair, while keeping the trial number of the other channel constant. We repeated this n = 200 times to create a null distribution of GC. The null distribution was exploited to calculate the percentile threshold P = 0.05.” We mark the frequency range of significant GC with a blue bar for encoding (dark blue spectrum exceeds light blue spectrum, information flow from cortex to hippocampus) and with a red bar for maintenance (dark red spectrum exceeds light red spectrum, information flow from hippocampus to cortex).

We now mention the null distribution also in Results (Line 170 – 172, 175 – 176). The net information flow ΔGranger (GC_hippcortex_ – GC_cortexhipp_) during encoding was significant in the 8-18 Hz range (blue bar in Figure 2 i, p<0.05 permutation test against a null distribution).

8) In Figure 2b – why EEG, and not ECOG, signal is being analyzed?

We now highlight the difference in recording modality in Results (Line 133 – 135).

9) I appreciate that authors present single subject results in Figure 4 but I believe group level analysis should be also performed to support conclusions.

Figure 5a shows a group level analysis of the single participant results of Figure 4. The panels of Figure 4 allow comparison across all participants.

We have created the new Table 2 to facilitate the understanding of how reliable the effects are across subjects.

10) Does the quality/magnitude of GC estimation depends on the number of trials? If sofor Figure 5b authors show subsample number of trials in the correct condition to match the number of trials in the incorrect condition.

We thank the reviewer for asking this question.

To balance trial numbers, we calculated the GC in a subset of correct trials (median of 200 permutations of a number of correct trials that equals the number of incorrect trials for each participant). The distributions between encoding and maintenance differed (P = 1e-4, paired cluster based permutation test, Figure 5 b). It thus turned out that the GC estimation does not deteriorate when balancing the numbers of correct and incorrect trials. We now present the balanced finding in the Results (Figure 5 b, Line 245 – 248).

11) Authors should state how many sessions were used in each comparison.

We have now created Table 1 that lists the number of correct trials for each participant.

12) In the statement:"The GC was lower than for ECoG, as expected for the lower signal amplitude of scalp EEG." I do not know what "lower" means – is it a magnitude or frequency of GC? If frequency, how does it relate to the scalp EEG amplitude?

We apologize for the unclear wording. The scalp EEG signal is smaller in amplitude than the ECoG signal. We therefore assume that derived quantities like the GC are smaller, too.

We have now reworded the phrase for clarity (Line 223).

13) Throughout the paper it seems that "ΔGranger" is used to describe a different thing.

We used the term ΔGranger throughout this paper to describe differences in directionality of the Granger. Subtracting the Granger spectrum of one direction (cortex hippocampus) from the other (hippocampus cortex) we aim at showing the bidirectionality of information flow by means of sign reversal. This approach is common in the literature, e.g. (Jenison 2014).

We assume that the Reviewer is referring to the label of the color bars. This is not the percentage of change but simply the difference of GC (where we use the unit % for better legibility). We have now removed the % from the label for clarity (Line 576).

References

Baddeley A. 2003. Working memory: looking back and looking forward. Nat Rev Neurosci; 4:829-839. doi 10.1038/nrn1201.

Boran E, Fedele T, Klaver P, Hilfiker P, Stieglitz L, Grunwald T, Sarnthein J. 2019. Persistent hippocampal neural firing and hippocampal-cortical coupling predict verbal working memory load. Sci Adv; 5:eaav3687. doi 10.1126/sciadv.aav3687.

Boran E, Fedele T, Steiner A, Hilfiker P, Stieglitz L, Grunwald T, Sarnthein J. 2020. Dataset of human medial temporal lobe neurons, scalp and intracranial EEG during a verbal working memory task. Sci Data; 7:30. doi 10.1038/s41597-020-0364-3.

Jenison RL. 2014. Directional Influence between the Human Amygdala and Orbitofrontal Cortex at the Time of Decision-Making. PloS one; 9:e109689. doi 10.1371/journal.pone.0109689.

Kumar S, Gander PE, Berger JI, Billig AJ, Nourski KV, Oya H, Kawasaki H, Howard MA, Griffiths TD. 2021. Oscillatory correlates of auditory working memory examined with human electrocorticography. Neuropsychologia; 150:107691. doi https://doi.org/10.1016/j.neuropsychologia.2020.107691.

Miceli S, Ness TV, Einevoll GT, Schubert D. 2017. Impedance Spectrum in Cortical Tissue: Implications for Propagation of LFP Signals on the Microscopic Level. eNeuro; 4. doi 10.1523/ENEURO.0291-16.2016.

Michels L, Moazami-Goudarzi M, Jeanmonod D, Sarnthein J. 2008. EEG α distinguishes between cuneal and precuneal activation in working memory. NeuroImage; 40:1296-1310. doi 10.1016/j.neuroimage.2007.12.048.

Nunez P. 1995. Neocortical dynamics and human EEG rhythms: Oxford University Press.

Solomon EA, Kragel JE, Sperling MR, Sharan A, Worrell G, Kucewicz M, Inman CS, Lega B, Davis KA, Stein JM, et al. 2017. Widespread theta synchrony and high-frequency desynchronization underlies enhanced cognition. Nat Commun; 8:1704. doi 10.1038/s41467-017-01763-2.

Trongnetrpunya A, Nandi B, Kang D, Kocsis B, Schroeder CE, Ding M. 2015. Assessing Granger Causality in Electrophysiological Data: Removing the Adverse Effects of Common Signals via Bipolar Derivations. Front Syst Neurosci; 9:189. doi 10.3389/fnsys.2015.00189.

von Stein A, Sarnthein J. 2000. Different frequencies for different scales of cortical integration: from local γ to long range α/theta synchronization. Int J Psychophysiol; 38:301-313. doi.

[Editors’ note: the authors resubmitted a revised version of the paper for consideration. What follows is the authors’ response to the second round of review.]

We found the approach to working memory based on effective connectivity between sensory cortex and hippocampus during working memory maintenance and retrieval intriguing and novel. All of the referees appreciated the efforts that have been made to improve the manuscript. The principal reason for rejection is the statistical interpretation of the data raised by all three referees, who are concerned whether the analyses allow robust inference about effects in the population. Referee 1 highlights differences between subjects in the frequency bands of effective connectivity. The effects are not robust to re-referencing recommended by referee 2, who does not consider a common reference adequate for this work. Referee 3 is concerned about multiple comparisons.Reviewer #1:I appreciate the efforts the authors have taken to improve the manuscript. The additional information does aid interpretation of the study. I still have a couple of remaining core issues to be sorted.

We thank the reviewer for this appreciation.

1) In the abstract, it states that information flow from auditory cortex to hippocampus peaked in the range of 8-18 Hz. In the newly inserted (and very helpful) table it seems that the range is anywhere from 4 to 21 Hz. Unless I am missing something, it appears that the authors are still making conclusions based on only Participant 1 (who showed significance in the 8-18 Hz range). I feel that this confusion runs throughout the manuscript and should be corrected – that is, any points where generalizations are made from a single subject that are not truly consistent across subjects should be corrected and the variability should be highlighted/discussed.2) Similarly, the frequency band of hippocampus > auditory cortex is often not congruent with the AC > hippocampus within subjects, so it is too simplistic to say this was reversed during maintenance. Moreover, this difference in frequencies within subjects should be discussed.

We have now completely revised our data analysis. The results addressed by the reviewer (Figures 4 and 5 of the previous version) are now less relevant and no longer reported in the manuscript.

Following the reviewers’ comments, we have now improved the quality of our data. In particular, we have now

used separate references for hippocampal LFP and ECoG recordings,determined the cortical sources of our EEG recordings by beamforming,applied cluster-based permutation tests on the population of participants.

We have then improved our GC analysis by down sampling the data to the Nyquist frequency. While new GC analysis confirmed the reversal of the net information flow between encoding and maintenance, the significant differences now appeared in theta band in all analyses.

We have now described the results of our improved GC analysis in the Results sections 2.4, 2.5 and 2.6. We have created several new figures and panels. We now present the GC spectra for Participant 1 (Figure 2h), for Participants 2,3 (Figure 3 d,i) and for the beamforming sources of all participants in Figure 4a. Significant net information flow (ΔGranger) appeared in the theta frequency range for all pairs of signals that we analysed. We therefore selected the 4-8 Hz band to test the reversal of ΔGranger in all participants individually (Figure 4d) and showed that the reversal appears for correct trials only (Figure 4f).

Reviewer #2:The authors have attempted to answer my concerns by performing a bipolar and Laplacian analysis to the data prior to the connectivity analysis. Unfortunately, this revealed that effects were not preserved after applying local re-referencing. I appreciate the authors' arguments that some effects have a large spatial structure that is removed by bipolar. However, the electrodes used by the authors are quite far apart and a priori I would think that a referencing across a large spatial structure should preserve true effects while removing spurious ones.Unfortunately, given simulation work as in the article I referred to in my original review (Trongnetrpunya, Nandi, Ding, et al., Frontiers in Systems Neuroscience), the false positive rate is unacceptably high when Granger and related methods are applied to data sharing a common reference. The current results therefore may be incorrect and cannot be interpreted in their present form.

To address the reviewer’s concern, we have revised major parts of our data analysis.

We have re-analyzed the LFP and ECoG data with two separate references:

1) We re-referenced the hippocampal LFP against the signal of a depth electrode contact in white matter.

2) We re-referenced the cortical ECoG against a different depth electro de contact.

The choice of two separate references for LFP and ECoG has been shown to avoid spurious functional connectivity estimates (Bastos and Schoffelen, 2015).

Our main findings remain unchanged, apart from minor changes in the topography and that GC now appears in the theta frequency band. This new analysis has corroborated our previous findings of

1) net information flow from auditory cortex to hippocampus during encoding.

2) net information flow from hippocampus towards auditory cortex during maintenance.

We have now described the new re-referencing scheme in the Methods Section 4.5 “Recording setup, re-referencing, and preprocessing”. We have adapted the results of our analysis throughout the manuscript.

The authors also apparently have cherry picked their data in Figures 4 and 5 by first taking channels that show a change in Granger, and then analyzing those changes across participants. It is like forcing the data to behave as we would like, another example of a flawed approach taken by the present manuscript. Given these technical/methodological issues, I cannot recommend the manuscript for publication in the present state.

We have certainly not cherry picked our data. Nevertheless, as stated above we have thoroughly re-analyzed our data following the reviewer’s suggestion. This extensive re- analysis corroborated the earlier findings with improved statistics. The results addressed by the reviewer (Figures 4 and 5 of the previous manuscript) are now less relevant and no longer reported in the manuscript.

We have now used beamforming to determine the cortical sources of our EEG data. We then analyzed the Granger causality (GC) between the cortical sources and the hippocampal local field potentials (LFP). The statistics was then performed with cluster based permutation tests of the effects in the subject population. We now describe the new analysis of the GC in the source space in the Methods Sections 4.8 and 4.9. We have created a new Figure 4, a new supplementary figure (Figure S1) and modified the Table 1. We have now described the new results of our analysis throughout the manuscript.

I do want to commend the authors for recording this dataset, and performing directed connectivity analysis, which is rare for these types of data. I would encourage the authors to think more deeply about how to assess directed functional connectivity robustly and reformulate their work with this in mind.

We thank the reviewer for this encouragement. We have now analyzed the directed functional connectivity robustly on the beamforming source level, applied cluster based permutation tests and thereby reformulated our work accordingly.

The new analyses have corroborated and improved our earlier results.

Reviewer #3:The authors have revised their manuscript and attempted to address all of the comments raised by myself and the other reviewers.

We thank the reviewer for this appreciation.

I have few remaining issues.If authors compute the difference against null distribution not between conditions (for example light blue and dark blue) in Figure 3d,i and in Figure 4 – How statement "dark blue spectrum exceeds light blue spectrum, information flow from cortex to hippocampus" is correct? It seems that, for each light colour result (hippocampus to cortex during encoding, cortex to hippocampus during maintenance) there should be a result of a statistical test and it seems none of these tests were performed. The caption of Figure 4 where authors use ΔGranger seems also incorrect as authors compute statistics from null distribution.

We performed the analysis exactly as the reviewer suggested. We are sorry if this was not clear. We do not compute statistics from a null distribution but from a null distribution of differences.

Following the reviewer’s comment, we have now clarified the description of statistics in the Methods section 4.10:

“To assess the significance of the difference of the Granger between different directions, we compared the difference of the true values to a null distribution of differences.

We recomputed GC after switching directions randomly across trials, while keeping the trial numbers for both channels constant. Then we computed the difference of GC for the two conditions. We repeated this n = 200 times to create a null distribution of differences. The null distribution was exploited to calculate the percentile threshold p = 0.05. In this way, we compare the difference of the dark and light spectra against a null distribution of differences”.

Unfortunately, I believe that issue with multiple comparisons was not addressed entirely. For instance, the analysis in the frequency domain does not use any correction for multiple comparisons. Because of that it is unknown if small differences observed in results are not effects of multiple comparisons.

Following the reviewers’ comments, we have now thoroughly reanalyzed our data.

We have now used beamforming to determine the cortical sources of our EEG data (Methods 4.8). We then analyzed the Granger causality (GC, Methods 4.9) between the cortical sources and the hippocampal local field potentials (LFP). The statistics was then performed with cluster based permutation tests of the effects in the subject population (Methods 4.10). This corrects for multiple comparisons (false-rate-discovery) (Maris and Oostenveld, 2007). We statistically compared the spectral GC in EEG sources between the two directions (hipp->cortex, cortex-> hipp) at the group level. We first selected a task period (encoding or maintenance). We then selected a cortical source and calculated the GC to hippocampus. For auditory cortex, the cluster-based permutation tests on the group level (Figure 4a) revealed

1) net information flow from auditory cortex to hippocampus during encoding,

2) net information flow from hippocampus towards auditory cortex during maintenance.

This was also true for theta GC to other cortical areas on the group level, but with less significance (Figure 4b,c).

On the level of individual participants, the reversal of the net information flow between encoding and maintenance was significant (paired permutation test) only to auditory cortex (Figure 4d). There were statistically significant differences among the cortical areas regarding net information flow both during encoding (p = 0.0009, Kruskal-Wallis test) and maintenance (p = 0.001, Kruskal-Wallis test). We confirmed that the net information flow in auditory cortex was significantly different from any other area (Dunn’s test, Bonferroni corrected).

Additionally for Figure 5 b and c analysis it seems that the comparison is not designed correctly. For figure 5b the data is picked using significant reversal of net GC (ΔGranger) – this is not done for Figure 5c analysis so it is hard to draw any conclusions from this contrast.

We have now completely revised our data analysis. The results addressed by the reviewer (Figures 4 and 5 of the previous version) are now less relevant and no longer reported in the manuscript.

We have now used beamforming to determine the cortical sources of our EEG data (Methods 4.8) and performed cluster-based permutation tests (Methods 4.10). We have thereby replaced the old Figures 4 and 5 with a new Figure 4.

About the group statistics issue. Figure 5a although presents average results does not show any statistics thus it is unknown if any of those information flows is significant. Figure 4 can give me an intuition that the result is present, but I have no idea about how significant it is on the group level. I believe that paper would benefit from the group statistics.

We have now completely revised our data analysis. The results addressed by the reviewer (Figures 4 and 5 of the previous version) are now less relevant and no longer reported in the manuscript.

We have now used beamforming to determine the cortical sources of our EEG data (Methods 4.8) and performed cluster-based permutation tests (Methods 4.10). We used group cluster based permutation with multiple comparison correction on the group level to determine the frequency ranges where the GC differs significantly between the two directions for the two task periods. We have now added bars in new Figure 4a to indicate the frequency ranges where spectral GC differs on the group level. As stated above, for auditory cortex the cluster-based permutation tests on the group level (Figure 4a) revealed

1) net information flow from auditory cortex to hippocampus during encoding,

2) net information flow from hippocampus towards auditory cortex during maintenance.

References

Bastos AM, Schoffelen JM (2015) A Tutorial Review of Functional Connectivity Analysis Methods and Their Interpretational Pitfalls. Front Syst Neurosci 9:175.

Maris E, Oostenveld R (2007) Nonparametric statistical testing of EEGand MEG-data. Journal of Neuroscience Methods 164:177-190.

[Editors’ note: further revisions were suggested prior to acceptance, as described below.]

The work provides new insights into the communication between sensory cortex and hippocampus during working memory.The revised work has been seen by all of the original referees who agree it is substantially improved. They have raised a number of further points about analysis and exposition we would like you to consider when preparing the final article for acceptance.Reviewer #2:I appreciate some of the additions that the authors have included. I do have further suggestions on the manuscript.

We thank the reviewer for this appreciation.

No details are provided for the scalp EEG system. How many electrodes? Was this an active system? Filtering at all? Many of these points are particularly relevant and important for interpreting the beamforming data. It says that the electrodes were aligned for each participant – how was electrode localization performed? Using what system – Polhemus?

We thank the Reviewer for pointing us to this omission.

We have now described the scalp EEG recordings in more detail in the Methods.

Section 4.3

“For scalp EEG recording, cup electrodes (Ag/AgCl) were placed on the scalp and filled with electrolyte gel (Signagel, Parker Laboratories, parkerlabs.com) to obtain an impedance < 5 kΩ.”

Section 4.4

“The scalp EEG electrodes were placed at the sites of the 10-20 system by experienced technicians and no further localization was performed. While the 10-20 standard is 21 scalp electrodes, in some patients some electrode sites stayed vacant to assure the sterility of the leads to the intracranial electrodes, resulting in a median of 17 scalp sites per patient.”

Section 4.5

“All recordings (LFP, ECoG, and scalp EEG) were performed with the Neuralynx ATLAS system (sampling rate 4000 Hz, 0.5-1000 Hz passband, Neuralynx, neuralynx.com).”

Section 4.8.

“To solve the forward problem we used a precomputed head model template and aligned the EEG electrodes of each participant to the scalp compartment of the model via interactive scaling, translation and rotation (ft_electrode_realign.m).”

Was all source reconstruction done with a template MRI? If so, wouldn't it have been appropriate to use each individual's MRI for better localisation, given that you surely had these available? You could then do the group averages, but at least the individual subject data would be more accurately localised.

The reviewer is right. We used precomputed head models based on a template MRI. We agree that subject-specific head models yield source localization that may be more accurate. However, there are several factors that affect the variability in spatial localization and the use of a template MRI is only one of them. In our study, the localization accuracy at the group level appeared sufficient to support our main finding: information flows from auditory cortex and hippocampus during encoding and reverses direction during replay of memory items. We envisage higher localization accuracy in future studies.

The granger causality analyses are reported for only three of the subjects between hippocampal LFP and ECoG data. Did the other 12 subjects not have any cortical electrodes then, even without parietal coverage, only hippocampal LFPs? This isn't at all clear from the descriptions in the manuscript and seems surprising that data would only be available from hippocampus, and no other temporal regions, even if parietal coverage was lacking.

Planning of the intracranial electrode placement is a clinical decision. It reflects where the epileptologists hypothesize the epileptic foci to be localized (Zijlmans et al., 2019). Since the presumed epileptic foci included the hippocampus in all patients, electrodes were placed in the hippocampus. Only in four patients, additional ECoG electrodes were placed because an epileptic focus in the cerebral cortex was considered. In three of these patients, the ECoG electrodes were placed on the left hemisphere and the data analysis had been presented in Figure 2 and Figure 3. In the fourth patient, the ECoG electrodes were placed on the right parietal cortex and had not been presented before. While the spectral power was comparable to that of Participants 1-3, there was no significant Granger information flow from or to the right cortical hemisphere. This adds to the evidence from the scalp EEG sources that the phonological loop recruits predominantly the left cortical hemisphere.

Following the Reviewer’s comment, we now renumbered the participants so the participant with right parietal electrodes gets number 4. We added the findings of Participant 4 in five panels to Figure 3 and to the Results.

Section 2.2

“Similarly in the electrode contacts on right parietal cortex of Participant 4 (Figure 3 k), the letter stimulus elicited γ activity and the maintenance period showed α activity (8-11 Hz, Figure 3 l).”

Section 2.4

“However, in Participant 4, the recordings from the right cortical hemisphere (Figure 3 k), did not show significant Granger causality between LFP and ECoG during task performance (Figure 3 n,o).”

We now elaborate on the rationale for electrode placement in Methods Section 4.2 and have added the reference (Zijlmans et al., 2019).

“Electrodes were placed according to the findings of the non-invasive presurgical evaluation where the epileptologists hypothesized the epileptic foci to be localized (Zijlmans et al., 2019). Since the presumed epileptic foci included the hippocampus in all patients, electrodes were placed in the hippocampus. In four patients, additional ECoG electrodes were placed on the cortex because an epileptic focus in the cerebral cortex was considered.”

It would be beneficial to include a display of all the contact locations for hippocampus, for transparency and interpretability purposes, along with (or perhaps instead) a supplement including all MNI coordinates of localised electrodes.

We thank the reviewer for this suggestion. Note that recordings from both left and right hippocampus showed significant Granger information flow to the left cortical hemisphere.

Following the Reviewer’s suggestion, we have created two new supplements. In the new Supplementary File 1 we list all MNI coordinates of localized electrodes for hippocampus. In the new Figure 1 —figure supplement 1 we show the coordinates of localized electrodes for hippocampus projected in a left hippocampal surface.

Further, we have now prepared a new graphical representation of the hippocampal electrode locations in Figure 1d and Figure 4g.

We have now clarified that several participants had recordings in the right hippocampus.

Results section 2.6:

“Interestingly, the pattern appeared with LFP recorded from the right hippocampus in several participants (Supplementary File 1).”

Reviewer #3:The authors have quantified patterns of information flow during a working memory task between cortex and hippocampus. They capitalize on a unique dataset where human intracranial data have been collected from the hippocampus, in parallel with human electrocorticography and/or electroencephalography. The authors use a statistical method to infer functional directed connectivity. This reveals information flowing from cortex to hippocampus during encoding but the reverse direction during maintenance. This study is quite unique in its combination of methodology with data that can precisely track the anatomical sources of neural computations during a memory task.Recommendations for the authors:In particular, the nice use of beamforming show the area specificity for the effects in auditory cortex. The volume conduction paragraph should take this into account, but should also mention that the PLV and Laplacian based referencing scheme eliminated the effects. There should be a discussion about the possible reasons as to why this was observed in the author's data analysis.

We thank Reviewer #3 for suggesting to expand our discussion on volume conduction. In following this comment, we have added a phrase in the Results section that mentions our analysis on the scalp electrode level that we had presented in a previous version of the manuscript. However, we fear that readers would be confused if we discuss why previous analyses – that are not detailed in the present manuscript – were criticized. We now do discuss the implications of the beamforming technique.

Following the Reviewer’s comments, we have now mentioned in the Results section 2.6:

“A similar GC pattern emerged when using the signals from left temporal scalp electrodes but was eliminated when using a Laplacian derivation.”

We restructured the discussion paragraph on volume conduction and added the phrase:

“On the level of scalp EEG analysis, we used beamforming as a source reconstruction technique [33] to characterize the primary neuronal generators that were localized specifically in left auditory cortex.”

On a small technical note, raw Granger should not be multiplied by 100%. Please revert back to the raw Granger values, as this would make the work more comparable to other studies, and these numerical units are meaningful. It is OK to express GC percentage differences in the cortex to hippocampus vs. hippocampus to cortex directions during the task modulation, however, to emphasize the changing direction patterns.

Following the Reviewer’s comment, we have now reverted to raw Granger values in the figures, the figure captions, and the text. At the same time, we have removed the leading zero from all decimal numbers <1 to improve legibility and to comply with the standard in psychology publications.

Relative power plots are not showing any decreases in power relative to fixation. How were the data baselined? They vary from 0 to 1. What about power decreases from fixation?

Following the Reviewer’s comment, we have now extended the color scale to negative values in all panels that report relative power.

Lines 200-206 when reporting the change in Granger using a permutation test, but no p-values are given.

We thank the reviewer for pointing this omission. As stated in the Methods Section 4.10, statistical significance is established at p<.05.

Following the Reviewer’s comment, we have now mentioned in lines 210-213 ”blue bar, p<.05,” and “red bar, p<.05”.

References

Baddeley A (2003) Working memory: looking back and looking forward. Nat Rev Neurosci 4:829-839.

Zijlmans M, Zweiphenning W, van Klink N (2019) Changing concepts in presurgical assessment for epilepsy surgery. Nature Reviews Neurology 15:594-606.